# Evaluation of polar stratospheric clouds in the global chemistry-climate model SOCOLv3.1 by comparison with CALIPSO spaceborne lidar measurements

Michael Steiner[1,*], Beiping Luo[1], Thomas Peter[1], Michael C. Pitts[2], and Andrea Stenke[1]

[1]Institute for Atmospheric and Climate Science, ETH Zurich, Switzerland
[*]now at Laboratory for Air Pollution / Environmental Technology, EMPA, Switzerland
[2]NASA Langley Research Center, Hampton, Virginia 23681, USA
**Correspondence:** Michael Steiner (michael.steiner@empa.ch)

**Abstract.** Polar Stratospheric Clouds (PSCs) contribute to catalytic ozone destruction by providing surfaces for the conversion of inert chlorine species into active forms and by denitrification. The latter describes the removal of $HNO_3$ from the stratosphere by sedimenting PSC particles, which hinders chlorine deactivation by the formation of reservoir species. Therefore, an accurate representation of PSCs in chemistry-climate models (CCMs) is of great importance to correctly simulate polar
ozone concentrations. Here, we evaluate PSCs as simulated by the CCM SOCOLv3.1 for the Antarctic winters 2006, 2007 and 2010 by comparison with backscatter measurements by CALIOP onboard the CALIPSO satellite. The year 2007 represents a typical Antarctic winter, while 2006 and 2010 are characterised by above- and below-average PSC occurrence. The model considers supercooled ternary solution (STS) droplets, nitric acid trihydrate (NAT) particles, water ice particles, and mixtures thereof. PSCs are parameterized in terms of temperature and partial pressures of $HNO_3$ and $H_2O$, assuming equilibrium be-
tween gas and particulate phase. The PSC scheme involves a set of prescribed microphysical parameters, namely ice number density, NAT particle radius and maximum NAT number density. In this study, we test and optimize the parameter settings by several sensitivity simulations. The choice of the value for the ice number density affects simulated optical properties and dehydration, while modifying the NAT parameters impacts stratospheric composition via $HNO_3$-uptake and denitrification. Depending on the NAT-parameters, reasonable denitrification can be modeled. However, its impact on ozone loss is minor.
Best agreement with the CALIOP optical properties and observed denitrification was for this case study found with the ice number density increased from the hitherto used value of 0.01 to 0.05 $cm^{-3}$ and the maximum NAT number density from $5 \times 10^{-4}$ to $1 \times 10^{-3}$ $cm^{-3}$. The NAT radius was kept at the original value of 5 μm. The new parametrization reflects the higher importance attributed to heterogeneous nucleation of ice and NAT particles following recent new data evaluations of the state-of-the-art CALIOP measurements. A cold temperature bias in the polar lower stratosphere results in an overestimated
PSC areal coverage in SOCOLv3.1 by up to 40%. Offsetting this cold bias by +3 K delays the onset of ozone depletion by about two weeks, which improves the agreement with observations. Furthermore, the occurrence of mountain-wave induced ice, as observed mainly over the Antarctic Peninsula, is continuously underestimated in the model due to the coarse model resolution (T42L39) and the fixed ice number density. Nevertheless, we find an overall good temporal and spatial agreement between modeled and observed PSC occurrence and composition. This work confirms previous studies that also simplified PSC

schemes, which avoid nucleation and growth calculations in sophisticated, but time-consuming microphysical process models, may achieve good approximations of fundamental properties of PSCs needed in CCMs.

## 1  Introduction

Although the occurrence of clouds in the wintertime polar stratosphere has been observed for a long time, their importance for stratospheric ozone depletion was only recognized after the discovery of the Antarctic ozone hole in the mid 1980s (Farman

et al., 1985). Stratospheric clouds composed of supercooled ternary solutions (STS, $H_2SO_4$-$HNO_3$-$H_2O$ mixtures), crystalline nitric acid trihydrate (NAT) and water ice provide surfaces, on which inert reservoir species like HCl and $ClONO_2$ are transformed into active forms (Solomon et al., 1986). The activated species then are responsible for springtime ozone depletion induced by catalytic cycles (Molina and Molina, 1987). While STS droplets are responsible for most of the chlorine activation (Portmann et al., 1996; Kirner et al., 2015; Nakajima et al., 2016, and references therein), solid particles can in addition strongly

affect the chemical composition of the stratosphere. Especially NAT particles can, under certain conditions, grow to large particles with diameters of up to 20 μm, so-called NAT-rocks (Fahey et al., 2001). Their number density is small (Biele et al., 2001), but due to their size they reach high settling velocities and by sedimentation remove reactive nitrogen from the stratosphere. This so-called denitrification contributes to ozone depletion by hindering the formation of inactive reservoir species (Salawitch et al., 1993).

While the formation of water ice requires extremely cold conditions in the dry stratosphere, $HNO_3$-containing particles already occur at higher temperatures (Hanson and Mauersberger, 1988), and hence much more frequently. In contrast to solid particles, there is no nucleation barrier for liquid STS droplets, which form upon uptake of $HNO_3$ and $H_2O$ from the gas-phase by binary $H_2SO_4$-$H_2O$ solution droplets (Carslaw et al., 1995). Depending on the presence or absence of heterogeneous nuclei, different pathways of PSC formation exist (e.g. Fig. 2 in Hoyle et al., 2013).

PSCs can be observed by ground-based lidar instruments (e.g. Biele et al., 2001; Simpson et al., 2005), in airborne campaigns (e.g. Fahey et al., 2001) or by space-borne satellites (e.g. Michelson Interferometer for Passive Atmospheric Soundings (MIPAS); Fischer and Oelhaf, 1996; Fischer et al., 2008). Since 2006 the Cloud-Aerosol Lidar with Orthogonal Polarization (CALIOP) on CALIPSO (Cloud-Aerosol Lidar and Infrared Pathfinder Satellite Observations) measures PSCs with high vertical resolution (Winker and Pelon, 2003; Winker et al., 2007, 2009; Pitts et al., 2018). CALIOP measures backscatter intensities

at 532 nm and 1064 nm wavelength, and additionally separates the 532 nm backscatter signal into parallel and perpendicular polarized components. The depolarization ratio is a measure of the particle shape and allows to distinguish between liquid (spherical) and solid (aspherical) particles. This makes CALIOP a very suitable tool for observing and classifying PSCs.

Due to their critical role in stratospheric chemistry, the representation of PSCs is indispensable for atmospheric chemistry models. However, the complexity of PSC schemes varies considerably between models. Some models primarily aim at mim-

icking the effects of PSCs on chemical composition and vertical re-distribution of $HNO_3$ and $H_2O$ rather than at exactly reproducing PSC compositions. The detailed PSC formation along different pathways, depending on the presence or absence of heterogeneous nuclei, is usually not taken into account in those models. This is no problem under many circumstances, e.g.

when chlorine activation is close to saturation in the middle of an Antarctic winter, but an accurate knowledge of the hetero-geneous reaction and denitrification rates is essential for a quantitative description of polar ozone chemistry under transitional
conditions, as they occur at winter onset or in late winter and early spring, or at the far edge of the vortex. Therefore, some models include PSCs in a more sophisticated manner and aim at correctly simulating nucleation, growth and sedimentation of the different PSC types as well as the detailed redistribution of $HNO_3$ and $H_2O$.

Simple parametrizations form NAT or ice instantaneously either at the saturation temperature, or at a certain supersaturation. Below the onset temperature of NAT or ice, excess matter of $HNO_3$ or $H_2O$ is directly transferred into the particulate phase,
assuming equilibrium. The particle size then depends on assumptions made about the number density distribution or vice versa. Examples for global chemistry models using such PSC parametrizations are SOCOLv3.1 (Stenke et al., 2013), LMDZrepro (Jourdain et al., 2008) or CCSRNIES (Akiyoshi et al., 2009). More complex PSC schemes allow deviations from thermody-namic equilibrium and explicitly simulate nucleation, growth and evaporation of particles, as in CLaMS (Tritscher et al., 2019) or WACCM/CARMA (Garcia et al., 2007; Wegner et al., 2012; Zhu et al., 2017a). As particle sedimentation is important for the
chemical composition of the stratosphere, it is included in all PSC schemes. The settling velocity is mainly dependent on parti-cle size, which is either described by a modal size distribution (e.g. SOCOL, LMDZrepro), size bins (e.g. WACCM/CARMA, EMAC (Khosrawi et al., 2018), BIRA (Daerden et al., 2007)) or as single representative particles in models with Lagrangian sedimentation schemes (e.g. SCLaMS, ATLAS (Wohltmann et al., 2010), SLIMCAT/TOMCAT (Feng et al., 2011)). A detailed overview over the representation of PSCs in global models and its evaluation can be found in Grooß et al. (2020, under review).
Different approaches have been used to investigate the performance of PSC schemes, ranging from the evaluation of bulk properties like PSC areal coverage or air volume covered by PSCs up to detailed assessments of PSC properties along sin-gle satellite orbits. In addition, the impact of PSCs on the chemical composition or chlorine activation can be evaluated by comparison with observations of certain chemical species. Tritscher et al. (2019) recently presented a detailed evaluation of PSCs in CLaMS, including optical properties, geographical PSC volume, along-orbit comparisons and influence on gas-phase
$HNO_3$ and $H_2O$. Simulations for the Arctic winter 2009/2010 and the Antarctic winter 2011 show good agreement with ob-servations. However, the simulated $HNO_3$-uptake in early winter was stronger than observed and the permanent redistribution of $HNO_3$ was underestimated. A new PSC model in WACCM/CARMA, taking into account detailed microphysical processes, was presented by Zhu et al. (2017b) and Zhu et al. (2017a). For the Antarctic winter 2010, they found the optical properties of the simulated PSCs to compare well with CALIOP-observations. Also observed denitrification was well reproduced by the
model. After implementing ice nucleation on NAT and vice versa, the model is now able to capture PSCs with small NAT particles and large number densities as well. Other studies focused mainly on the impact of PSCs by comparing $HNO_3$ and $H_2O$ with space-borne observations from MLS (Microwave Limb Sounder; Waters et al., 2006; Schoeberl, 2007), MIPAS or with airborne measurements. The study by Khosrawi et al. (2018), evaluating EMAC for the Arctic winters 2009/2010 and 2010/2011, found good agreement for the temporal evolution of gas-phase $HNO_3$ in the polar stratosphere, but simulated PSC
volumes were smaller than observed by MIPAS. Recently, Snels et al. (2019) presented a statistical comparison including sev-eral models from CCMVal-2 and CCMI project with observations. They used a set of diagnostics, based on spatial distribution of ice and NAT surface area densities and temperature, to compare simulated PSCs among the different CCMs. They concluded

that the geographical distribution of PSCs in the polar vortex, as observed by CALIOP, is not well reproduced by the models. The models showed limited ability to reproduce the longitudinal variations in PSC occurrences and mostly overestimate NAT and ice occurrence, most probably due to a cold temperature bias. WACCM-CCMI (Garcia et al., 2017), where the cold bias was reduced by introducing additional mechanical forcing of the circulation via parametrized gravity waves, compared best with observations.

In this study, we compare a simple equilibrium scheme of STS, NAT, ice and mixtures thereof with state-of-the-art PSC satellite data, aiming to optimize the scheme for economic and efficient use in a chemistry-climate model (CCM). To this end, we evaluate the representation of PSCs in the CCM SOCOLv3.1 for the Antarctic winters 2006, 2007 and 2010. We convert the simulated PSCs into an optical signal to mimic a satellite measurement and compare the results with CALIPSO observations. We further evaluate the impacts of the simulated PSCs on the chemical composition of the stratosphere by comparison with MLS-satellite observations of $HNO_3$, $H_2O$ and $O_3$. A more detailed description of our methodology and the datasets utilized is given in Sect. 2. In Sect. 3 we present the results of the comparison, and Sect. 4 provides conclusions.

## 2 Model description and observational data

### 2.1 The SOCOLv3.1 chemistry-climate model

The state-of-the-art chemistry-climate model SOCOLv3.1 (Stenke et al., 2013; Revell et al., 2015) is based on the middle-atmosphere general circulation model (GCM) MA-ECHAM5 (European Centre/HAMburg climate model; Roeckner et al., 2006), coupled to the chemistry module MEZON (Model for Evaluation of oZONe trends; Egorova et al., 2003). MEZON contains 57 chemical species, 56 photolysis reactions, 184 gas-phase reactions and 16 heterogeneous reactions in and on aqueous sulfuric acid aerosols (supercooled binary solutions, SBS) as well as three types of PSCs, namely STS droplets, NAT and water ice. The chemistry module MEZON covers stratospheric ozone chemistry in detail as well as the tropospheric background chemistry, including the oxidation of isoprene (Pöschl et al., 2000). The coupling between the GCM and the chemistry module takes place through simulated winds and temperatures, as well as through the radiative forcing caused by ozone, methane, nitrous oxide, water vapor and CFCs. The dynamical time step is 15 min, whereas the radiation and chemistry schemes are called every 2 h.

In SOCOLv3.1, STS droplets form upon the uptake of gas-phase $HNO_3$ and $H_2O$ by aqueous sulfuric acid aerosols, following the expression by Carslaw et al. (1995). The binary aerosols are prescribed from a monthly mean observational data record, mainly based on SAGE (Stratospheric Aerosol and Gas Experiment) observations. This data set was prepared for CMIP6 (Eyring et al., 2016), and provides surface area density (SAD), volume density, mean radius and $H_2SO_4$ mass of the binary aerosol. The uptake of $HNO_3$ and $H_2O$ leads to a change in aerosol mass, from which a growth factor for the SBS particles and, therefore, the STS particle size is calculated. The stratospheric aerosol data set and its description can be found at ftp://iacftp.ethz.ch/pub_read/luo/CMIP6/.

NAT is formed if the $HNO_3$ partial pressure exceeds its saturation pressure (Hanson and Mauersberger, 1988). For NAT particles, we fix the mean radius and limit the maximum number density. The latter accounts for the fact that NAT and STS

clouds are mostly observed simultaneously (e.g., Pitts et al., 2011), and prevents condensation of all available gas-phase $HNO_3$ onto NAT particles at the expense of STS formation. In the reference set-up, we assumed a mean radius ($r_{NAT}$) of 5 μm and a maximum number density ($n_{NAT,max}$) of $5 \cdot 10^{-4}$ cm$^{-3}$ (Table 1). These settings allow for $\sim$10% of the $HNO_3$ at beginning of winter to be taken up into NAT particles (0.77 ppbv at 50 hPa and 195 K, assuming 5 ppmv $H_2O$).

For water ice, we prescribe the particle number density ($n_{ice}$). The reference setting of 0.01 cm$^{-3}$ represents background conditions, but not ice clouds formed due to mountain waves, where very high nucleation rates result in much higher ice number densities of $\sim$ 5-10 cm$^{-3}$ (Hu et al., 2002) and particle sizes of $<$3 μm (Höpfner et al., 2006). As for STS droplets the PSC routine assumes the water ice particles to be in thermodynamic equilibrium with the gas-phase.

The different treatment of NAT and water ice in the SOCOL model is motivated by the respective timescales to reach

equilibrium between particulate and gas-phase. For water ice, this timescale is very short. Once ice has formed, further cooling leads rather to particle growth than to additional nucleation of fresh particles. In case of NAT, however, the equilibrium between particulate and gas-phase is hardly reached, as shown by observations (e.g., Fahey et al., 2001), and additional particles can nucleate upon further cooling.

Sedimentation of solid PSC particles is included in the model. The fall velocities of NAT and ice particles are based on

Stokes theory (described in Pruppacher and Klett, 1997). NAT and ice PSCs are not explicitly transported in SOCOL. At the end of the chemistry routine all condensed $HNO_3$ and $H_2O$ evaporates back to the gas phase. This means that at each call of the chemistry routine NAT and ice PSCs (re-)form instantaneously depending on the prevailing partial pressures of $HNO_3$ and $H_2O$, respectively. This approach avoids undesired numerical diffusion due to the spatial heterogeneity in PSC occurrence. To prevent spurious PSC formation caused by potential model temperature, $HNO_3$ and/or $H_2O$ biases in regions where PSCs

are usually not observed, and to avoid overlap with the regular cloud scheme of the GCM, the occurrence of PSCs is spatially restricted. Water ice particles are allowed to occur between 130 hPa and 11 hPa and polewards of 50°N/S. NAT particles are allowed between the tropopause and 11 hPa. STS and NAT particles may form at all latitudes.

For the present study SOCOLv3.1 was run with T42 horizontal resolution (about 2.8°x 2.8°in latitude and longitude) and 39 vertical levels between the surface and the model top centered at 0.01 hPa ($\sim$80 km). In order to allow for a direct com-

parison with observations, the model was run in specified dynamics mode, i.e. the prognostic variables temperature, vorticity, divergence and the logarithm of the surface pressure are relaxed towards ERA-Interim reanalysis data (Dee et al., 2011). We applied a uniform nudging strength throughout the whole model domain, with a relaxation timescale of 24 h for temperature and logarithm of the surface pressure, 48 h for divergence and 6 h for vorticity. The boundary conditions follow the specifications of the reference simulation REF-C1 of phase 1 of the Chemistry Climate Model Initiative (CCMI-1; Morgenstern et al.,

2017). All simulations for this study were run for the time period from 1 May to 31 October with a 12-hourly output time step. We chose the years 2006, 2007 and 2010 for our evaluation. While 2007 represents a typical Antarctic year with a steady vortex and PSCs observed from May to September, 2006 and 2010 are years with above- and below-average PSC occurrence, respectively. All years are without volcanic influence.

## 2.2 CALIPSO PSC observations

The simulated PSCs in SOCOL are compared to measurements from the CALIOP instrument onboard CALIPSO, an Earth observation satellite in the A-train constellation in operation since 2006 (Winker and Pelon, 2003; Winker et al., 2007, 2009). The A-train of satellites orbits the Earth 14-15 times per day, covering the latitudes between 82°S and 82°N on each orbit. CALIOP is a dual-wavelength lidar with three receiver channels, one measuring the 1064 nm backscatter intensity, the two others measuring the parallel and perpendicular polarized components ($\beta_\parallel$ and $\beta_\perp$) of the 532 nm backscattered signal. The frequency of the lidar pulse is 20.25 Hz, corresponding to one measurement every 333 m along the flight track. From the measured backscatter coefficients (e.g. $\beta_{532}$) the total (sum of particulate and molecular) to molecular backscatter ratio

$$R_{532} = \frac{\beta_{532}}{\beta_\mathrm{m}} = \frac{\beta_{\mathrm{part},532} + \beta_\mathrm{m}}{\beta_\mathrm{m}} \tag{1}$$

can be calculated, with $\beta_\mathrm{m}$ being the molecular backscatter coefficient. $\beta_\mathrm{m}$ is calculated as described in Hostetler et al. (2006), using molecular number density profiles provided by the MERRA-2 (Modern-Era Retrospective analysis for Research and Applications, version 2) reanalysis products (Gelaro et al., 2017). With the separation of the 532 nm backscatter signal into parallel and perpendicular polarized components, the depolarization ratio $\delta_\mathrm{aerosol}$ (perpendicular to parallel component) of the 532 nm signal can be derived, which is an indicator of the particle shape and hence phase (liquid/solid).

In this study, we use the Lidar Level 2 Polar Stratospheric Cloud Mask Product (available via Michael C. Pitts), which was derived with version 2 (v2) of the PSC detection algorithm (Pitts et al., 2018) from the CALIOP v4.10 Lidar Level 1B data products. This CALIOP PSC dataset contains profiles of PSCs with classification and optical properties, also providing temperature, pressure and tropopause height derived from MERRA-2 reanalyses. The spatial resolution of PSC data is 5 km in the horizontal by 180 m in the vertical. Only night-time measurements are considered.

Version 2 of the detection algorithm (Pitts et al., 2018) detects PSCs as statistical outliers in either $\beta_\perp$ or $R_{532}$, relative to the background stratospheric aerosols population. The optical properties of stratospheric background aerosol are derived from CALIOP measurements above 200 K. Both thresholds are defined as median plus one median absolute deviation. The values are calculated on a daily basis and vary with potential temperature. Furthermore, additional horizontal averaging (over 15, 45 and 135 km) has been implemented into the PSC detection algorithm to enable the detection of more tenuous clouds than at 5 km resolution only.

The PSC classification in Pitts et al. (2018) distinguishes STS, STS-NAT mixtures, enhanced NAT mixtures, ice and wave ice. The categories are visualized in Fig. 1. The dotted lines denote dynamical boundaries, while the solid lines show boundaries at fixed $\beta_\perp$ or $R_{532}$ values. The lines at the lower left corner approximate the $\beta_\perp$-threshold ($\beta_{\perp,\mathrm{thresh}}$) and $R_{532}$-threshold ($R_{532,\mathrm{thresh}}$), respectively. All PSCs above $\beta_{\perp,\mathrm{thresh}}$ are assumed to contain non-spherical particles. The boundary between the two NAT mixture categories and ice is calculated "dynamically", i.e. based on cloud-free MLS measurements of $HNO_3$ and $H_2O$. PSCs are detected as wave ice when they contain non-spherical particles and if $R_{532} > 50$. A detailed description of the classification scheme is given in Pitts et al. (2018). PSC observations of July 2007 (Fig. 1) show the most distinct relative maxima for STS. Two further relative maxima appear with higher $\delta_\mathrm{aerosol}$ values, indicating solid particles. The relative maximum extending towards the upper left corner of the histogram corresponds to STS-NAT mixtures with low NAT number

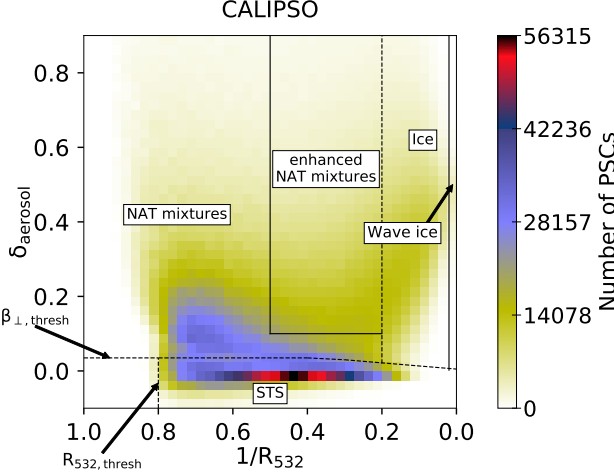

**Figure 1.** Composite 2D-histogram of CALIPSO PSC measurements of July 2007 in a $1/R_{532}$-$\delta_{\text{aerosol}}$ coordinate system with 40x40 bins. The colors indicate the number of PSC measurements in one bin. Dotted lines denote dynamical classification boundaries or thresholds and solid lines denote fixed classification boundaries.

densities ($n_{\text{NAT}}$), while the second relative maximum extending towards the upper right corresponds to mixtures of NAT with high number densities and ice as well as to wave ice PSCs.

## 2.3 MLS observations

In this study, modeled $HNO_3$, $H_2O$ and $O_3$ mixing ratios are compared to satellite measurements of the Microwave Limb Sounder (MLS) onboard the Aura satellite (Waters et al., 2006). MLS measures atmospheric profiles by scanning from the ground to 90 km height in flight direction, passively measuring microwave thermal emissions. All three quantities are derived by version 4.2 from the Aura MLS Level 2 data (Livesey et al., 2018). The $HNO_3$ dataset has a vertical resolution of approximately 3-4 km, while the $H_2O$ and $O_3$ datasets have a vertical resolution of 2.5 to 3 km. The accuracy of the MLS measurements is 1-2 ppbv for $HNO_3$ (Santee et al., 2007), 4%-7% for $H_2O$ (Read et al., 2007; Lambert et al., 2007) and 8% for stratospheric $O_3$ (Jiang et al., 2007). Detailed informations and a precise description of the dataset can be found in Livesey et al. (2018).

## 2.4 Model-measurement comparison

While CALIOP measures backscatter signals and depolarization ratios, the SOCOL model provides surface area densities for STS, NAT and water ice as function of pressure, latitude and longitude. From the outputted SADs of the three PSC types and the prescribed microphysical parameters, i.e. $r_{NAT}$ and $n_{ice}$, as well as the growth factor for liquid aerosols we calculate the number density and/or radius for each particle type. These quantities are used in Mie and T-matrix scattering codes (Mishchenko et al., 1996) to compute optical parameters of the simulated PSCs, i.e. $R_{532}$, $\delta_{\text{aerosol}}$ and $\beta_{\perp}$, for comparison

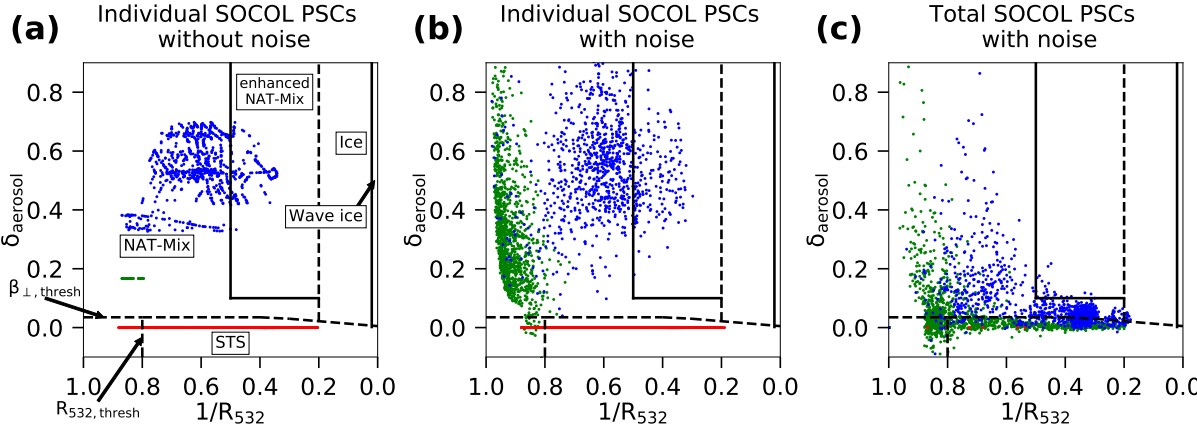

**Figure 2.** Scatter plot of simulated PSCs from the SOCOL simulation $S_{REF}$ on 1 July 2007 in a $1/R_{532}$-$\delta_{\text{aerosol}}$ coordinate system. (a): STS (red), NAT (green) and ice (blue) as individual components. (b): As in (a), but after applying observational uncertainties. (c): The modeled PSCs as mixture of all components present per grid box (red: pure STS, green: STS-NAT mixtures, blue: mixtures with ice) with uncertainty.

with CALIOP observations. For NAT and ice particles, circular symmetric spheroids with an aspect ratio of 0.9 are assumed. Refractive indices of 1.31 for water ice and 1.48 for NAT (Middlebrook et al., 1994) were chosen. STS are liquid particles and therefore assumed to be spherical, which corresponds to a depolarization ratio $\delta_{\text{aerosol}} = 0$.

The CALIOP PSC data product includes detection thresholds, $R_{532,\text{thresh}}$ and $\beta_{\perp,\text{thresh}}$, for each measurement. As the geographical PSC extent strongly depends on these detection limits, they have to be applied to the calculated optical properties of

the simulated PSCs as well to ensure a fair comparison between model and satellite data. For this purpose, we calculated for each pressure level the daily mean thresholds over all observations.

The satellite measurements are subject to uncertainties. Even for a perfectly monodisperse PSC distbution a CALIPSO measurement would show some scatter. To ensure best possible comparability between model and measurements, observational uncertainties have to be applied to the calculated optical properties of the modeled PSCs. We followed the approach by Engel

et al. (2013). The uncertainty scales inversely to the square root of the horizontal averaging distance along a flight path, which we set to 135 km. This value corresponds to the best case for detection, which maximizes the comparability with the model (which obviously does not have a detection threshold). An example for the added measurement noise is shown in Fig. 2. When looking at the three PSC types individually (Fig. 2a), STS (due to their assumed spherical shape) and NAT (due to the fixed radius) appear at constant $\delta_{\text{aerosol}}$-values of 0 and 0.167, respectively. The variable radius of ice particles results in a variable

$\delta_{\text{aerosol}}$-value. Applying the uncertainties to the parallel and the perpendicular backscatter coefficients primarily causes a large spread in depolarization ratio (Fig. 2b). When considering all PSC particles to be mixed within a grid box (Fig. 2c), they appear mainly at the lower and left side of the composite histogram.

## 3 Results and discussion

Since our results and conclusions do not substantially differ for the three analyzed winters, we focus here on the year 2007, a typical Antarctic winter. Figures for the winters 2006 and 2010 are shown in the Appendix (Fig. A3-A14). We start with the analysis of our reference simulation (Table 1). The sensitivity simulations are discussed in Sect. 3.4.

### 3.1 Comparison along an orbit

As a first example, we compare SOCOL with CALIPSO along a single flight track. Figure 3 shows a curtain of observed inverse backscatter ratios $1/R_{532}$ along orbit 2 on 1 July 2007 (Fig. 3a) and the corresponding PSC compositions (Fig. 3d). The observations indicate a large PSC over the Antarctic Peninsula (270 - 300°E), and a smaller PSC over Oates Land (160 - 190°E). Further, some tropospheric cirrus clouds were classified as PSCs. Above the Antarctic Peninsula, two distinctive regions with very small $1/R_{532}$ values are evident. These high backscatter ratios ($R_{532} > 50$) are related to high number densities of ice particles (up to 10 cm$^{-3}$, Hu et al., 2002), which are caused by rapid cooling rates associated with mountain wave events. These wave ice clouds are surrounded by more synoptic scale PSCs with lower $R_{532}$ values, which are classified as ice, STS and NAT mixtures.

Figures 3b and 3d show the corresponding plots for the PSCs as simulated by the SOCOL model in the respective grid boxes overflown by CALIPSO. Figures 3c and 3f show the same, but before detection thresholds and instrument uncertainty had been added. The model output also reveals a large PSC over the Antarctic Peninsula. However, the spatial extent of the simulated PSC is larger. The simulated backscatter ratio $R_{532}$ peaks around 6, which is substantially lower than observed. Due to the coarse resolution and the rather smooth orography in the model, SOCOL is not able to capture high ice particle number densities associated with small-scale mountain wave events. Applying the CALIOP classification scheme on the model output results in a layer of ice PSCs located around ~20 km, which is slightly higher than in the observations. The ice cloud is surrounded by NAT mixtures, while the observations indicate STS. Below those NAT mixtures, pure STS clouds occur in the model (Fig. 3f), most of which are tenuous enough such that they fully disappear after applying the optical thresholds (Fig. 3e).

The actual modeled composition (Fig. A1) shows a similar pattern than the CALIOP classification scheme, but with more ice Mix and less STS. This difference between actual composition and the composition according to the CALIPSO classification scheme of SOCOL PSCs can also be seen in Fig. 2c, where most of the ice mixtures (blue) are located in the NAT-Mix domain, while many NAT mixtures (green) are located in the STS domain. It should be noted that the modeled optical properties are exclusively calculated for PSCs. Tropospheric cirrus clouds treated by the model's cloud routine are therefore excluded.

### 3.2 Spatial distribution

Figure 4 presents monthly mean (including clear-sky and cloudy-sky conditions) backscatter ratios $R_{532}$ from observations and simulation for July (a and b) and August 2007 (c and d). For a better comparison, the high-resolution measurements have been gridded onto the SOCOL grid. The data are vertically averaged over all pressure levels above the tropopause. The observations show a month-to-month variability in the location of the PSC region. In July, the mean backscatter intensity appears to be more

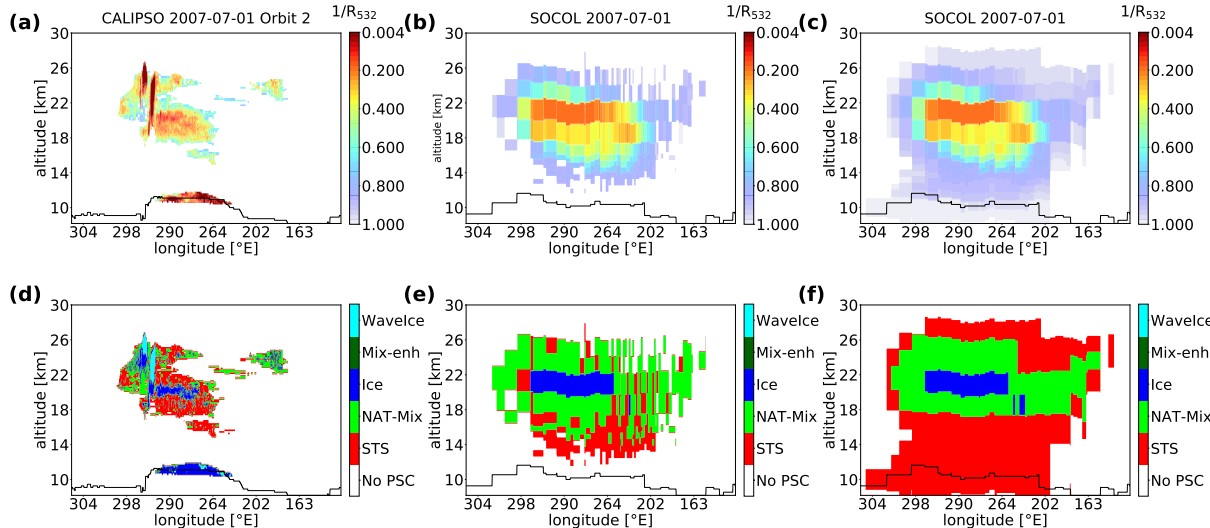

**Figure 3.** CALIPSO measurements on 1 July 2007 (orbit 2) of $R_{532}$ (a) and the PSC classification (d). Calculated $R_{532}$ values for modeled PSCs from the simulation $S_{REF}$ in the overflown grid boxes after adding the instrument uncertainty and applying the detection thresholds are shown in (b). (e) shows the composition of the corresponding PSCs according to the classification scheme in Pitts et al. (2018). (c and f): The same as in (b and e), but without instrument uncertainty and the detection thresholds. The black lines indicate the WMO and model tropopause height for CALIPSO measurements and simulations, respectively.

homogeneously distributed, with a slight peak over East Antarctica ($\sim$0-150$°$E), while in August a distinct peak downstream of the Antarctic Peninsula ($\sim$55-70$°$W) is observed. This characteristic feature is caused by frequent mountain wave events in this region (Hoffmann et al., 2017). These mountain waves lead to the formation of wave ice with very high backscatter values, but also to subsequent formation of enhanced NAT-mix clouds with high number densities of NAT particles (Zhu et al., 2017a).

The modeled month-to-month variability in the $R_{532}$ values and areal extent agrees well with CALIPSO observations. In

July, the center of the PSC area is also tilted towards East Antarctica and slightly towards the Peninsula in August. However, peak values of $R_{532}$ are clearly lower for SOCOL, and the spatial distribution is more homogeneous. As mentioned above, this results mainly from a poor representation of mountain waves in the model, but also from the fixed ice number density and upper limit for the NAT number density. Although the years 2006 and 2010 show a slightly different seasonal cycle (Fig. A5, A6), the conclusions on the model performance hold for those years as well.

## 3.3 PSC areal coverage

The total areal coverage as a function of altitude and time is a measure for the seasonal evolution of PSCs inside the polar vortex. Figure 5 compares CALIOP observations and model results for the winter 2007 (see also Fig. 13 in Pitts et al., 2018). The modeled PSC area is determined for every grid box based on the PSC occurrence for two output time steps per day, 0:00 and 12:00 UTC. We consider the entire model grid box to be covered by PSCs as soon as PSCs occur and exceed the detection

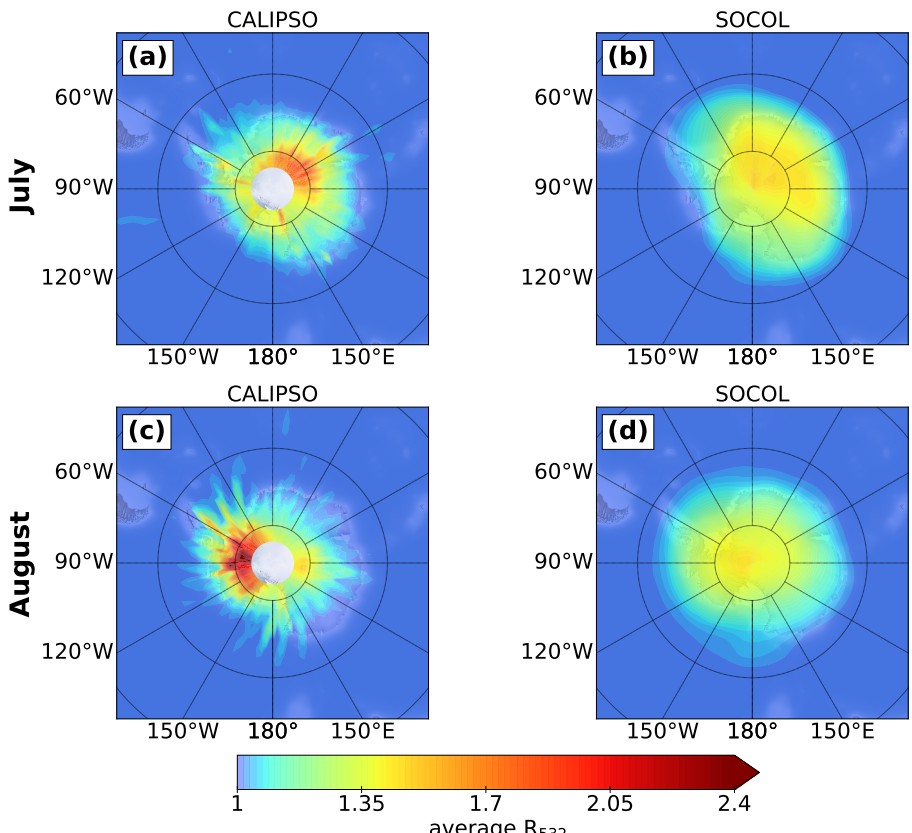

**Figure 4.** Vertically integrated monthly means of $R_{532}$ for all-sky conditions as observed by CALIPSO (a and c, gridded onto the SOCOL grid) and as simulated by SOCOL from the simulation $S_{REF}$ (b and d), for July (top) and August 2007 (bottom).

limits. The observed PSC area is calculated in two different ways: (1) from the daily fraction of PSC measurements within ten equal-sized latitude bands as described in Pitts et al. (2018) (Fig. 5a), and (2) from measurements average over 12 hours and gridded onto the SOCOL grid (Fig. 5b). The second method is similar to the calculation of the modeled PSC area ("SOCOL-method"). Since CALIOP does not over-pass all SOCOL grid cells within 12 hours, the "empty" grid cells are filled by the PSC area in the over-passed grid cells at the same latitude. We applied the "SOCOL-method" to the CALIOP data to achieve the best possible comparability between model and observations. Compared to Fig. 5a, the "SOCOL-method" leads to an increase in the CALIOP PSC areal coverage. Since CALIOP does not provide data poleward of 82°, measurements between 77.8 and 82°S are assumed to be representative of the entire 77.8–90°S latitude band.

Considering the low-level (11 - 12 km) clouds in May and June as tropospheric cirrus, the first PSC occurrence is observed in mid-May at 20-25 km altitude (Fig. 5a). Periods with higher PSC areal coverage with large vertical extent alternate with periods of less PSC extent. A clear peak occurs at end of July between 17 and 23 km altitude. The PSC areal coverage starts to decrease beginning of September, reaching zero mid-October. The descent of the coldest temperatures within the winter

season is reflected in the descent of PSC occurrence. As described in Pitts et al. (2018), the PSCs merge with tropospheric cirrus clouds at mid-July.

In SOCOL, PSC formation starts about 2 weeks earlier (Fig. 5c). The model is capable of reproducing the temporal occurrence of the individual peaks end of July. Also the overall descent of maximum PSC coverage is present in the simulation. PSCs exist until end of October, which is longer than observed. Furthermore, SOCOL simulates a substantially larger PSC area than observed (Fig. 5a), in particular between 13 and 23 km altitude, where $1.5 \cdot 10^7$ km$^2$ are almost continuously exceeded.

There are two main reasons for the overestimated PSC area and for the longer PSC period in the model. Part of the overestimation can be explained by the calculation method, since even small amounts of PSCs within a large model grid cell contribute substantially to the PSC areal coverage. However, SOCOL still overestimates CALIPSO when applying the "SOCOL-method" to the observations (Fig. 5b). Most of the overestimation results from a cold temperature bias in the polar lower stratosphere, which is typically around 2 to 4 K. Offsetting this cold bias by +3 K in a sensitivity simulation results in a decrease in the simulated PSC areal coverage (Fig. 5e) and a clearly improved agreement with CALIOP observations (Fig. 5b).

The modeled PSC area calculated without the optical thresholds applied (Fig. 5d and f) is significantly larger, especially below 13 km altitude, where large areas with STS clouds occur in the model (see also Fig. 3f). Those large-scale STS clouds are very tenuous and filtered out by applying the conservative PSC detection threshold. The contribution of those STS clouds to SAD is negligible. However, the comparison highlights the crucial role of the detection thresholds for model-measurement intercomparisons. Due to this sensitivity to the applied methods, quantitative comparisons of the areal coverage must be interpreted with caution.

Observed and simulated PSC coverage for the years 2006 and 2010 are shown in Figs. A5 and A6. In 2006, the year with above-average PSC occurrence, offsetting the cold bias leads to a smaller PSC coverage than observed, indicating that not the synoptic-scale temperature, but rather small-scale temperature fluctuations determined the PSC occurrence and areal coverage in 2006 (see also Fig. A3). As such small-scale features are not adequately represented in SOCOL, correcting for the synoptic-scale temperature bias leads to an underestimation of the PSC coverage.

## 3.4 Sensitivity to microphysical parameters

As described in Sect. 2.1, SOCOL's PSC scheme includes some prescribed microphysical parameters such as the ice particle number density, $n_{ice}$, or the NAT radius, $r_{NAT}$. These values had once been chosen based on what was known about PSCs back then. However, the current parameter setting might not be optimal. For example, the rather low value for $n_{ice}$ of 0.01 cm$^{-3}$ prevents the formation of ice PSCs with high number densities as observed in mountain wave events. To investigate the sensitivity of the simulated PSCs to the parameter setting, we performed additional simulations with increased $n_{ice}$ and/or increased $n_{NAT,max}$ (Table 1). In addition, we performed a simulation with increased temperatures for PSC formation to investigate the effect of the cold temperature bias on simulated PSCs and chemical composition within the polar vortex.

Figure 6 shows the composite histograms for various SOCOL simulations. There are considerable differences to the observations (Fig. 1), but also between the simulations. PSCs in the reference simulation $S_{REF}$ show a strong relative maximum located in the STS domain with $1/R_{532}$ values between 0.4 and 0.2 (Fig. 6a). Only very few PSCs are classified as ice, i.e. the

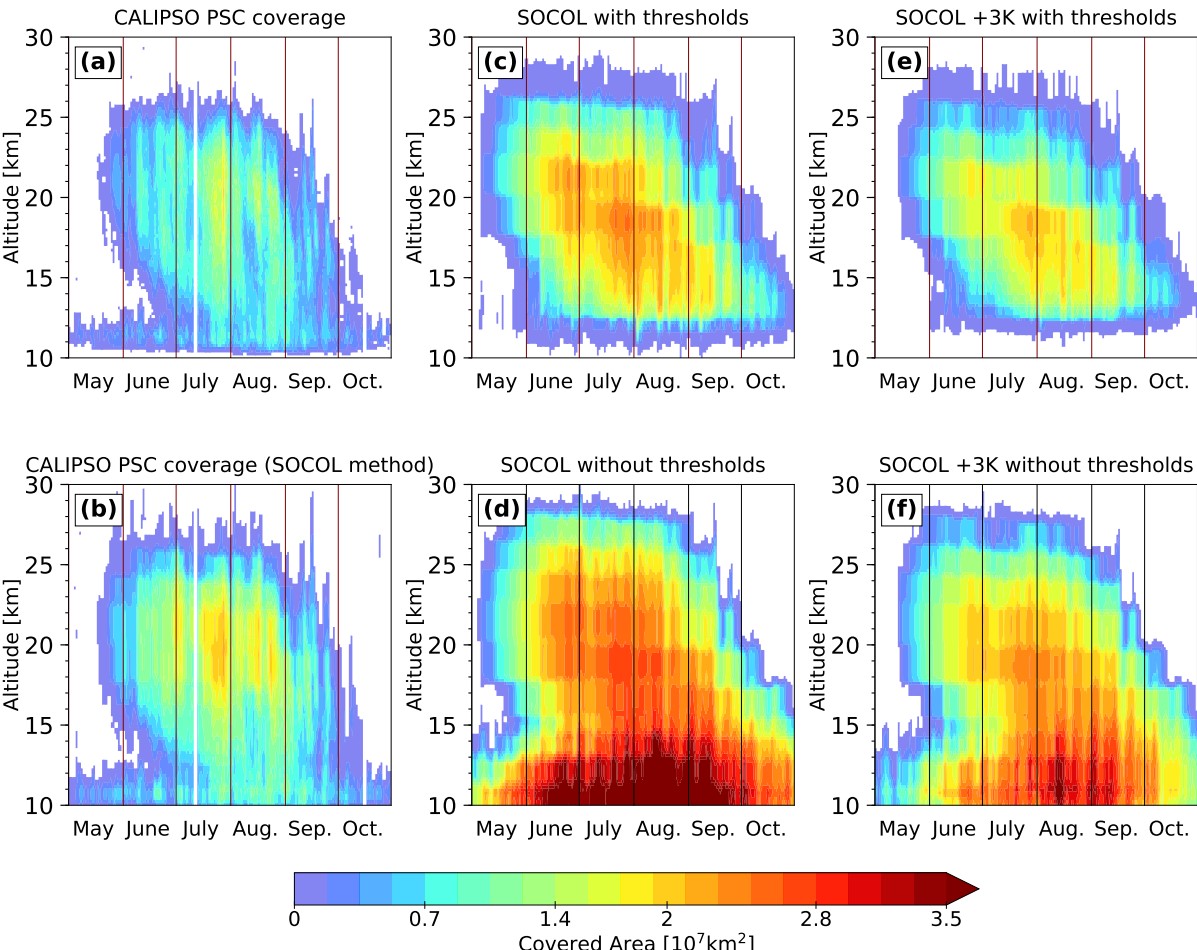

**Figure 5.** Time series of total PSC areal coverage over the Antarctic region as a function of altitude for the winter 2007. (a) derived from CALIOP as described in Pitts et al. (2018). (b) derived from CALIOP by applying the "SOCOL-method" (see text). (c) and (d): derived from the SOCOL reference simulation with and without applying detection limits and instrument uncertainty, respectively. (e) and (f): same as for (c) and (d), but with PSC formation temperature increased by 3 K.

relative maximum towards the upper right, as observed by CALIPSO, is missing. The shift of modeled PSC towards the lower and left side of the histogram is also visible in Fig. 2c. There are several reasons for this difference: First, SOCOL does not resolve small-scale mountain waves due to the coarse horizontal resolution and smooth orography applied in the model. Furthermore, the modeled PSCs are representative for large grid box (2.8°x2.8° horizontally and approximately 2 km vertically), while the observations resolve much smaller scale structures (starting from 5 km horizontally along a track and 180 m vertically). Finally, the fixed ice number density of 0.01 $cm^{-3}$ and upper limit for NAT number densities do not allow for large ice and NAT cross sections, even if mountain waves would be resolved. Based on these findings we performed one sensitivity simulation with a tenfold ice number density, $S_{n(ice)}$. As shown Fig. 6b the tenfold increase in $n_{ice}$ results in a strong maximum


**Table 1.** Overview over the SOCOL simulations and the microphysical parameter settings.

| Parameter | $n_{\text{ice}}$ | $n_{\text{NAT,max}}$ | $r_{\text{NAT}}$ | $\text{T}_{PSC-formation}$ |
|---|---|---|---|---|
| $\text{S}_{REF}$ | 0.01 cm$^{-3}$ | 5x10$^{-4}$ cm$^{-3}$ | 5 $\mu$m | |
| $\text{S}_{n(ice)}$ | **0.1 cm$^{-3}$** | 5x10$^{-4}$ cm$^{-3}$ | 5 $\mu$m | |
| $\text{S}_{n(NAT,max)}$ | 0.01 cm$^{-3}$ | **2x10$^{-3}$ cm$^{-3}$** | 5 $\mu$m | |
| $\text{S}_{n(ice),n(NAT,max)}$ | **0.05 cm$^{-3}$** | **1x10$^{-3}$ cm$^{-3}$** | 5 $\mu$m | |
| $\text{S}_{T,n(ice),n(NAT,max)}$ | **0.05 cm$^{-3}$** | **1x10$^{-3}$ cm$^{-3}$** | 5 $\mu$m | +3 K |

towards the upper right, mainly within the enhanced NAT mixture domain. The higher number density of ice particles increases
the cross section of ice, leading to enhanced backscatter in ice-containing grid cells. Due to its solid state, backscatter from ice has $\delta_{\text{aerosol}}>0$. This results in a shift towards higher $R_{532}$ and higher $\delta_{\text{aerosol}}$ values in the histogram. Overall, modifying $n_{\text{ice}}$ leads to a better agreement in optical properties with CALIPSO.

NAT PSCs play a twofold role in stratospheric ozone chemistry: Besides efficient halogen activation on their surfaces, the sedimentation of NAT particles leads to denitrification, which hinders deactivation of reactive halogens and facilitates catalytic
ozone loss (Peter, 1997). While ice PSCs are less important for stratospheric ozone chemistry, NAT formation and subsequent denitrification of the stratosphere play a crucial role. NAT formation in SOCOL depends on two parameters, $n_{\text{NAT,max}}$ and $r_{\text{NAT}}$. To test the model's sensitivity to those parameters, we ran further simulations with the upper boundary for NAT number densities increased by a factor of four, $\text{S}_{n(NAT,max)}$, and the NAT radius increased from 5 to 7 µm. As both simulations showed similar changes, the latter is not presented here.

The simulation with four times higher $n_{\text{NAT,max}}$ (Fig. 6c) shows a maximum shifted towards lower $R_{532}$ values compared to the REF simulation, which is located around the optical thresholds at the lower left corner. As long as temperatures are below $T_{\text{NAT}}$ and enough HNO$_3$ is available for NAT formation, an increase in $n_{\text{NAT,max}}$ or $r_{\text{NAT}}$ results in more HNO$_3$-uptake by NAT particles. This reduces the available gas-phase HNO$_3$ for STS growth. Also, more HNO$_3$ through sedimentation of the solid NAT particles is removed. With larger $r_{\text{NAT}}$ this removal occurs even faster due to the higher sedimentation velocity.
The reduction in surface area density of STS results in less backscatter and subsequently a shift towards lower $R_{532}$ values in the composite histogram. This shift towards lower $R_{532}$ values worsens agreement with observations.

In a further simulation ($\text{S}_{n(ice),n(NAT,max)}$, Fig. 6d) we set $n_{\text{ice}}$ to 0.05 cm$^{-3}$ and $n_{\text{NAT,max}}$ to $10^{-3}$ cm$^{-3}$. This simulation shows a superposition of the two effects described above, resulting in two distinct relative maxima in the composite histogram. One maxima is located to the upper right, similar to $\text{S}_{n(ice)}$. The second maximum at low $R_{532}$ and low $\delta_{\text{aerosol}}$ values is similar
to the pattern in $\text{S}_{n(NAT,max)}$. The shift towards lower $R_{532}$ values is again a result of less STS formation due to the reduced availability of HNO$_3$. Although the composition histograms of all sensitivity simulations still differ substantially from the observations, we find the best agreement for the simulation $\text{S}_{n(ice),n(NAT,max)}$. Similar shifts in the composite plots between the various model simulations as discussed above can be found for 2006 and 2010 (Figs. A7 and A8). In the model simulation

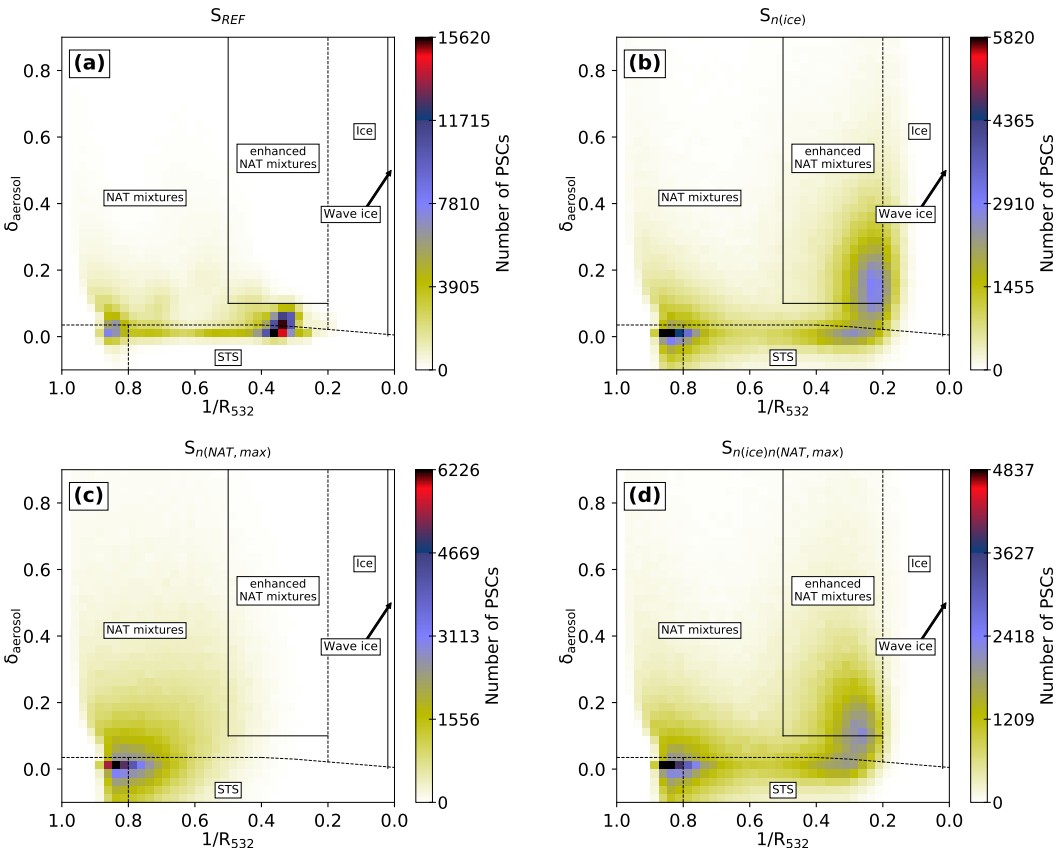

**Figure 6.** Composite 2D-histograms for July 2007, analogue to Fig. 1, for the simulations $S_{REF}$ (a), $S_{n(ice)}$ (b), $S_{n(NAT,max)}$ (c) and $S_{n(ice),n(NAT,max)}$ (d).

$S_{T,n(ice),n(NAT,max)}$ including a cold bias correction of +3 K (Fig. A2) the synoptic-scale temperatures are too warm for
substantial ice formation, emphasizing the importance of small-scale temperature fluctuations for ice PSCs.

To investigate the impact of the applied modifications on the simulated chemical composition of the polar stratosphere (60–
82°S), we compare modeled gas-phase $HNO_3$, $H_2O$ and $O_3$ with MLS measurements for 46 and 68 hPa (Figs. 7 - 9). To
account for the spatial heterogeneity of the MLS measurements, we first averaged the measurements over the SOCOL grid
boxes. Afterwards we calculated area-weighted polar mean concentrations. The top panels shows absolute values for MLS and
the $S_{REF}$ simulation, while the lower panels show the temporal evolution for MLS and all model simulations relative to 1 May.

At the beginning of winter, all simulations have similar $HNO_3$ concentrations, which are about 20% to 50% lower than MLS,
depending on the pressure level (Fig. 7). At 46 hPa MLS $HNO_3$ starts to decline around mid-May and in early June at 68 hPa.
Prior to the decline, an increase in $HNO_3$ is observed at 68 hPa. This results from the evaporation of sedimenting NAT particles
formed at higher altitudes (renitrification) and is an indication of denitrification of the upper levels. During July/August the
absolute $HNO_3$ values from the reference run $S_{REF}$ agree well with the observations. However, in late winter SOCOL again

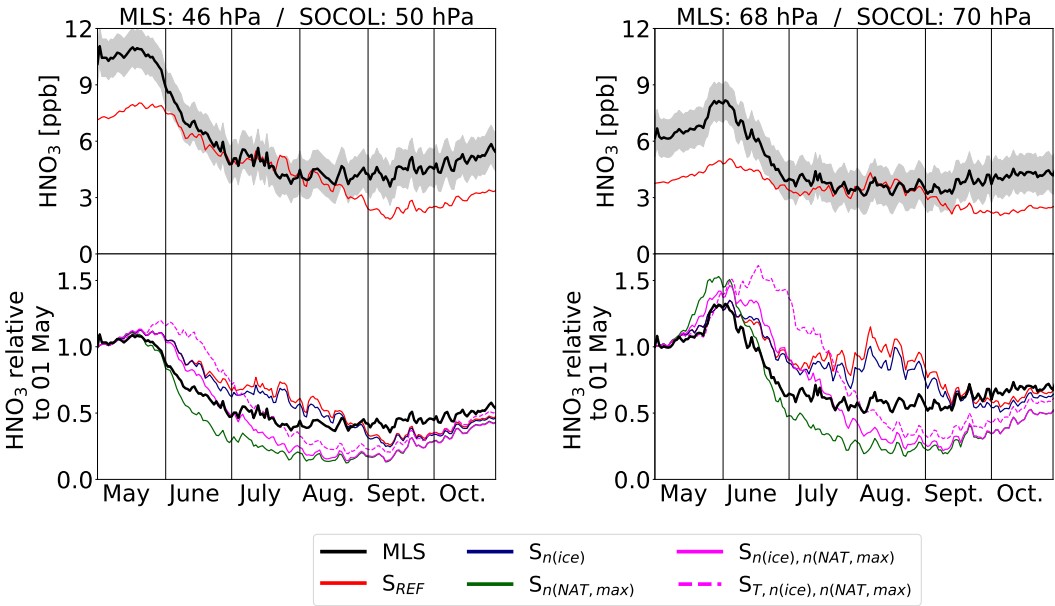

**Figure 7.** Temporal evolution of polar (60°S-82°S) mean gas-phase $HNO_3$ from MLS measurements and the different model simulations The uncertainty-range (gray shading) represent the MLS accuracy.

underestimates $HNO_3$ compared to MLS. All simulations show a decline due to $HNO_3$-uptake into NAT particles and STS droplets. However, $S_{REF}$ (red) and $S_{n(ice)}$ (dark-blue) show a weaker and delayed $HNO_3$ decline with a plateau in July/August.

In $S_{n(NAT,max)}$ (green) the decline at both levels is considerably stronger than in $S_{REF}$ as well as in MLS. This is due to the enhanced uptake of $HNO_3$ into NAT particles and the subsequent removal by sedimentation. As a consequence also the renitrification at lower levels is clearly enhanced. Both indicates a more efficient denitrification than in $S_{REF}$.

The simulation $S_{n(ice),n(NAT,max)}$ (magenta), in which $n_{NAT,max}$ is twice as large as in $S_{REF}$, but only half of $S_{n(NAT,max)}$, falls in between the other simulations. The denitrification starts about half a month later than in $S_{n(NAT,max)}$. The $HNO_3$-uptake is reduced and subsequently $HNO_3$ stays longer in the gas-phase. However, in August $HNO_3$ concentrations reach about the same level as in $S_{n(NAT,max)}$. Simulations with enhanced $r_{NAT}$ have similar effects (not shown).

In $S_{T,n(ice),n(NAT,max)}$ denitrification as well as renitrification are delayed by about half a month due to the later onset of PSC formation. However, towards the end of the winter, $HNO_3$ concentrations are almost the same in all model simulations.

Figure 8 shows the same as Fig. 7, but for $H_2O$. As for $HNO_3$, all simulations start with similar $H_2O$ values in May, but underestimate MLS by 20% to 30%. At 46 hPa MLS $H_2O$ starts to decline beginning of June. Rehydration of lower levels due to the evaporation of sedimenting ice particles is observed shortly after. At 68 hPa, MLS $H_2O$ starts to decrease mid of June. All model simulations except for $S_{T,n(ice),n(NAT,max)}$ show a very similar temporal evolution of $H_2O$ in the polar stratosphere and a very good agreement with MLS. In SOCOL the amount of ice is determined by the amount of available $H_2O$ and temperatures. The smaller the chosen $n_{ice}$, the larger the ice particles and the stronger the dehydration due to faster

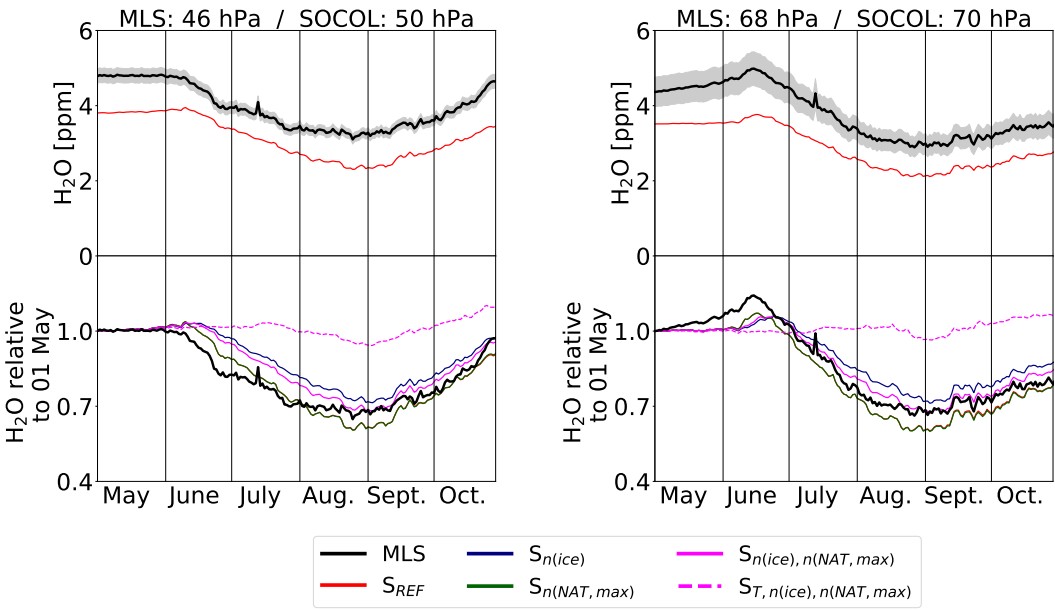

**Figure 8.** Same as Fig. 7, but for $H_2O$. Note that the line of $S_{n(NAT,max)}$ overlays $S_{REF}$, since these simulations have identical $H_2O$.

sedimentation. $S_{REF}$ and $S_{n(NAT,max)}$, the simulations with the lowest $n_{ice}$ of 0.01 cm$^{-3}$, show the strongest dehydration and the earliest onset, while $S_{n(ice)}$ with $n_{ice} = 0.1$ cm$^{-3}$ shows the smallest dehydration of the simulations without modified

PSC formation temperature. With the cold bias correction of +3 K, almost no dehydration takes place due to lack of ice formation. Changes in polar vortex $H_2O$ from modifying $n_{ice}$ have an influence on the SAD of NAT and STS, with higher $H_2O$ concentrations leading to larger NAT and STS SADs. However, this effect is small compared to the effects from modifying the NAT-related microphysical parameters, and therefore, not further discussed.

Finally, Fig. 9 presents simulated $O_3$ in the polar stratosphere compared to MLS. At the beginning of winter all model

simulations are in very good agreement with MLS measurements. For both pressure levels, the simulations show an earlier and stronger decline in $O_3$ than observed by MLS. Also, the recovery of $O_3$ starts earlier, leading to slightly higher $O_3$ values at the end of October. The spread among the model simulations is small compared to the differences to the observations, indicating minor effects of the PSC parameters on $O_3$-depletion. Increasing the parameter $n_{ice}$ slightly reduces the simulated dehydration, but the increased SAD of ice leads to a slightly stronger $O_3$ depletion in $S_{n(ice)}$ compared to $S_{REF}$. Allowing for higher

NAT number densities overall reduces SAD of PSCs due to reducing the abundance of $HNO_3$. However, due to enhanced denitrification, $S_{n(NAT,max)}$ and $S_{n(ice),n(NAT,max)}$ show even slightly lower $O_3$ concentrations. $O_3$-depletion starts later in $S_{T,n(ice),n(NAT,max)}$ due to the later onset of PSC occurrence and smaller PSC area. However, from end of August onwards the differences between the individual model simulations vanish. The discussed findings for $HNO_3$, $H_2O$ and $O_3$ hold also for the years 2006 and 2010, as shown in Figs. A11 to A14.

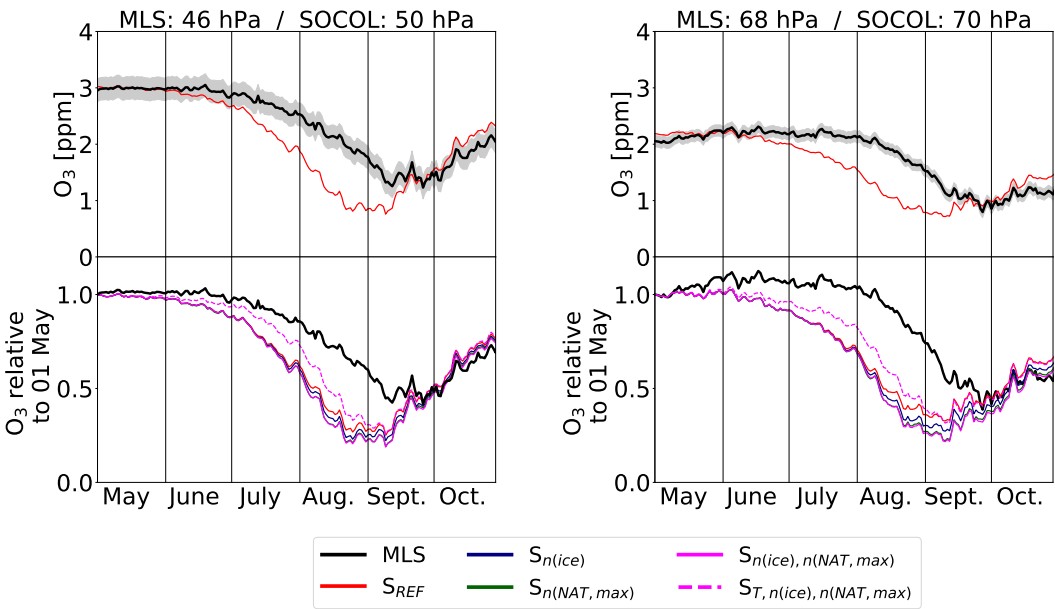

**Figure 9.** Same as Fig. 7, but for $O_3$.

## 4 Discussion and Conclusions

We presented an evaluation of polar stratospheric clouds (PSCs) in the Antarctic stratosphere as simulated by the chemistry-climate model SOCOLv3.1. The model was nudged towards ERA-Interim reanalysis (specified dynamics mode). We compared modeled PSC occurrence and composition to CALIPSO/CALIOP satellite observations by mimicking a lidar measurement on the model output. The impact of PSCs on the chemical composition of the polar stratosphere by denitrification, dehydration and ozone depletion was investigated by comparison with Aura/MLS satellite data. We analysed three winters with different PSC occurrence: 2006 (above-average), 2007 (average) and 2010 (below-average).

SOCOL considers STS droplets as well as water ice and NAT particles. PSCs are parametrized in terms of temperature and partial pressures of $HNO_3$ and $H_2O$, assuming equilibrium conditions. NAT and ice PSCs form instantaneously without taking the history of the air mass and preexisting PSC distributions into account. This approach states a simplification, but is successfully applied in other models as well (e.g., Wegner et al., 2013).

The PSCs scheme includes several fixed microphysical parameters, namely the maximum NAT number density, NAT radius and ice number density. Fixing the NAT radius leads to a homogeneous sedimentation velocity for all NAT particles, but allows for varying NAT number densities. Other models choose the reverse approach with fixed number densities, which results in varying NAT radius and sedimentation velocities (e.g., Wegner et al., 2013; Nakajima et al., 2016). In reality, the actual value for the NAT number density is far from being constant, because the active sites for NAT nucleation themselves show a

wide distribution of efficacies (Hoyle et al., 2013). Both approaches require some tuning of the microphysical parameters to reasonably simulate observed sedimentation and denitrification. This was done here by various sensitivity simulations.

Overall, the spatial distribution of modeled PSCs is in reasonable agreement with CALIOP observations, and the model captures the observed month-to-month variability. However, due to the coarse model resolution and mean orography, but also due to the fixed ice number densities and upper limit for NAT number densities, mountain-wave induced ice and enhanced NAT clouds with high backscatter ratios, mainly observed over the Antarctic Peninsula, are not resolved by the SOCOL model. The PSC areal coverage over Antarctica indicates a continuous overestimation of PSCs in SOCOL. As shown by a sensitivity simulation, this can be partly explained by the simulated cold temperature bias in the winter polar stratosphere, which prevails despite running the model in specified dynamics mode. Furthermore, it is a consequence of the large grid size: even a small amount of PSCs within a grid cell adds a large contribution to the areal coverage.

The choice of the prescribed ice number density, $n_{ice}$, determines primarily the optical signal and dehydration of the polar vortex through its impact on the particle size and, therefore, sedimentation velocities. While increasing $n_{ice}$ from the original value of $0.01$ cm$^{-3}$ to $0.1$ cm$^{-3}$ improves the agreement of the optical signal with CALIOP measurements, the simulated dehydration is more realistic for smaller $n_{ice}$ and, therefore, larger ice particles.

The upper limit for NAT number densities determines the HNO$_3$-uptake and subsequently the competition between simulated NAT and STS formation, which is crucial for halogen activation. Increasing the maximum NAT number densities improves the temporal agreement of de- and renitrification with MLS measurements. However, SOCOL clearly underestimates observed HNO$_3$ in the polar stratosphere already before the PSC season, which makes a solid conclusion about the best set of microphysical parameters difficult. Despite stratospheric H$_2$O and in particular HNO$_3$ being very sensitive to changes in the microphysical parameters, we found the impact on O$_3$ depletion to be surprisingly small.

Eliminating the cold temperature bias inside the polar vortex has a more pronounced impact on O$_3$ concentrations. The onset of O$_3$ depletion is delayed by one to two weeks. However, the maximum O$_3$ decline in September is overestimated by all model simulations compared the MLS. This suggests either a too strong heterogeneous ozone loss in the SOCOL model or shortcomings regarding the model's dynamics inside the polar vortex. The latter was discussed by Khosrawi et al. (2017) as potential reason for the underestimated polar vortex ozone concentrations in the EMAC chemistry-climate model. Brühl et al. (2007) found that, even in specified dynamics mode, the downward transport in the lower part of the polar vortex is too weak. Since the SOCOL model is based on the same general circulation model as EMAC, the underestimated polar stratospheric ozone concentrations in SOCOL are not necessarily exclusively caused by too strong chemical ozone destruction, but could also be related to a too weak downward transport, diminishing the re-supply with ozone-rich air masses from higher altitudes. This would explain why polar stratospheric ozone in the SOCOL model showed to be rather insensitive to modifications in the PSC scheme.

The co-existence of NAT and STS poses a substantial challenge to PSC parameterizations. As mentioned above, the SOCOL model addresses this issue by setting an upper limit for NAT number densities. Khosrawi et al. (2017, 2018) found underestimated PSC volume densities and denitrification/re-nitrification in the Arctic polar vortex simulated by the chemistry-climate model EMAC. The authors explained these findings with an unrealistic partitioning of gasphase HNO$_3$ into STS and NAT, with

NAT forming first at the expense of STS, the main contributor to PSC volume density. In addition, the simulated NAT particles may be too small for significant gravitational settling and re-nitrification of lower atmospheric levels. Khosrawi et al. (2018) suggested an adjusted $HNO_3$ partitioning and/or an upper limit for NAT number densities, as applied in the SOCOL model, as one potential way to improve the model. A similar approach was implemented by Wegner et al. (2013) in the WACCM model. To account for the simultaneous occurrence of STS and NAT, they allow 20% of total available $HNO_3$ to form NAT at a supersaturation of 10, with a NAT number density of $10^{-2}$ cm$^{-3}$. This value is an order of magnitude larger than the upper limit applied in SOCOL. An even larger NAT number density of $10^{-1}$ cm$^{-3}$ was used by Nakajima et al. (2016) in the ATLAS model. As Wegner et al. (2013), they allowed 20% of $HNO_3$ to go into NAT, while the rest is available for STS. These examples demonstrate that the best parameter setting for PSC schemes is strongly model dependent.

For the present study, we ran the model in a rather coarse resolution of T42L39. While higher resolutions are often discussed to improve the model performance, we do not expect any substantial drawbacks from the applied set-up. First, the model was used in specified dynamics mode, and we do not see major differences in modeled polar vortex temperatures or dynamics between a T42L39 and T42L90 simulation in nudged mode. Second, to capture mountain wave events, for example, we would need to go to very high horizontal resolutions, which are beyond the capabilities of current chemistry-climate models. This is supported by Khosrawi et al. (2017), who found only little differences in modeled polar stratospheric $HNO_3$ and $O_3$ between a T42 and a T106 horizontal resolution. Even with an anticipated resolution of T255 (60 km or 0.54°at the equator) they would expect problems with the representation of small-scale temperature fluctuations due to mountain waves. To account for the effects of mountain-waves on PSC formation Orr et al. (2015) implemented a parameterization of stratospheric temperature fluctuations into the global chemistry-climate configuration of the UK MetOffice Unified Model. They found an increase of up to 50% in PSC SAD over the Antarctic peninsula during early winter. Despite the fact that the SOCOL model experiences a cold temperature bias in the polar winter stratosphere, the Antarctic peninsula is indeed a region with relatively too little PSC occurrence in the model (Fig. 4). This underestimation is even more pronounced in our sensitivity runs with increased PSC formation temperature. In a very recent study Orr et al. (2020) showed that the additional mountain-wave induced cooling leads to enhanced NAT SAD throughout the winter and beginning of spring, resulting in intensified chlorine activation, especially during late winter/early spring. Interestingly, the effects of the parameterized mountain-wave cooling are not limited to the Antarctic peninsula, but involve the whole polar vortex. These findings emphasize the important role of ozone for atmospheric dynamics and the climate system.

In summary, we found the best overall agreement with the CALIOP and MLS measurements with the NAT and ice number concentrations increased from their default values to $n_{ice} = 0.05$ cm$^{-3}$ and $n_{NAT,max} = 1 \cdot 10^{-3}$ cm$^{-3}$, respectively. Our findings hold for all analyzed Antarctic winters. Further work would be required to extend our findings to simulated PSCs in the Arctic, which shows a more pronounced interannual variability than Antarctica. Our study confirms previous studies showing that also simplified PSC schemes based on equilibrium assumptions may achieve good approximations of fundamental properties of PSCs. However, the best parameter set-up is strongly model dependent. General model deficiencies like temperature biases or transport influence the parameter choice, and should be prioritized in future model development.

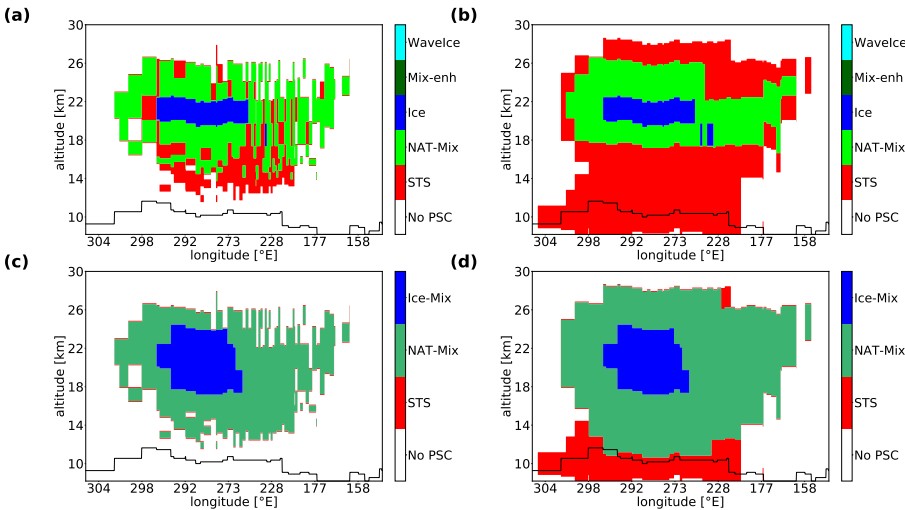

**Figure A1.** Composition of simulated PSCs along the CALIPSO orbit 2 on 1 July 2007 according to the classification scheme from Pitts et al. (2018) after (a) and before (b) applying detection limits and instrument uncertainty. Panels c) and d) show the modeled PSC type (STS: STS occurrence only; NAT-Mix: NAT but no ice occurrence; Ice-Mix: ice occurrence).

*Code and data availability.* Since the full SOCOLv3.1 code is based on ECHAM5, users must first sign the ECHAM5 license agreement before accessing the SOCOLv3.1 code (http://www.mpimet.mpg.de/en/science/models/license/, last access: 2020). Then the SOCOLv3.1 code is freely available. The contact information for the full SOCOLv3.1 code as well as the source code of the PSC module and the Mie and T-matrix scattering code are available at http://doi.org/10.5281/zenodo.4094663. The simulation data presented in this paper can be downloaded from the ETH Research Collection via http://dx.doi.org/10.3929/ethz-b-000406548. CALIPSO lidar level 2 polar stratospheric
cloud mask version 2.0 (v2) is available on request to Michael C. Pitts. MLS $HNO_3$, $H_2O$ and $O_3$ data products have been downloaded from https://mls.jpl.nasa.gov/index-eos-mls.php (latest access 1.11.2018).

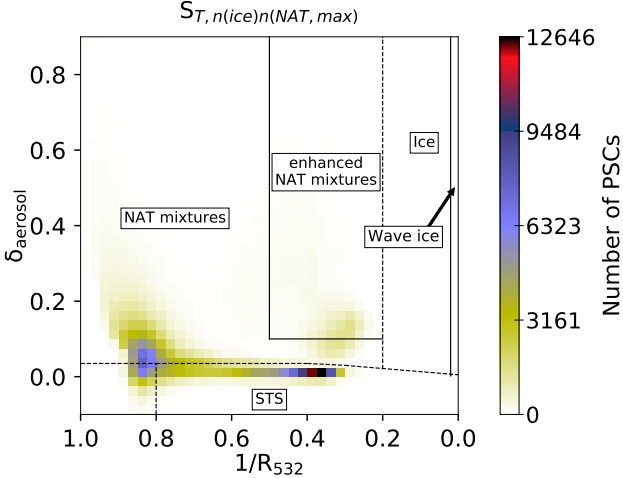

**Figure A2.** Composite 2D-histogram for July 2007, analogue to Fig. 1, for the simulation $S_{T,n(ice),n(NAT,max)}$.

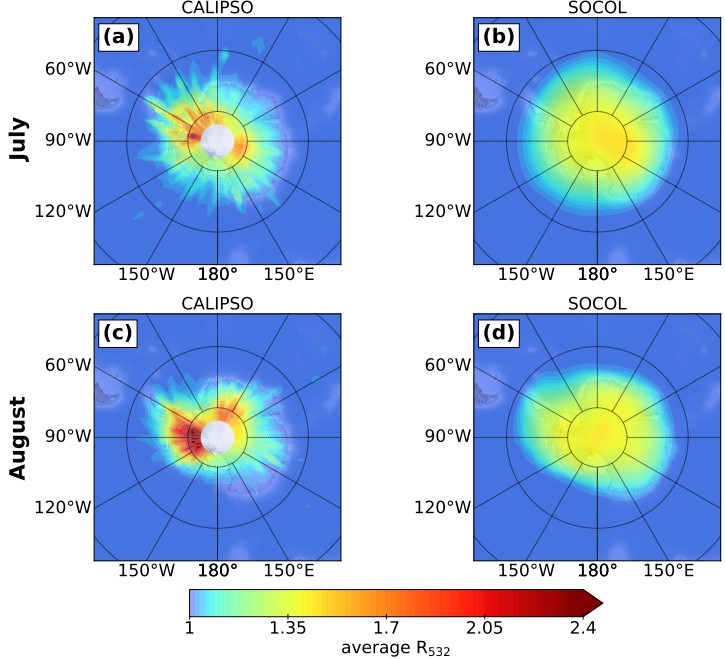

**Figure A3.** As Fig. 4, but for the year 2006.

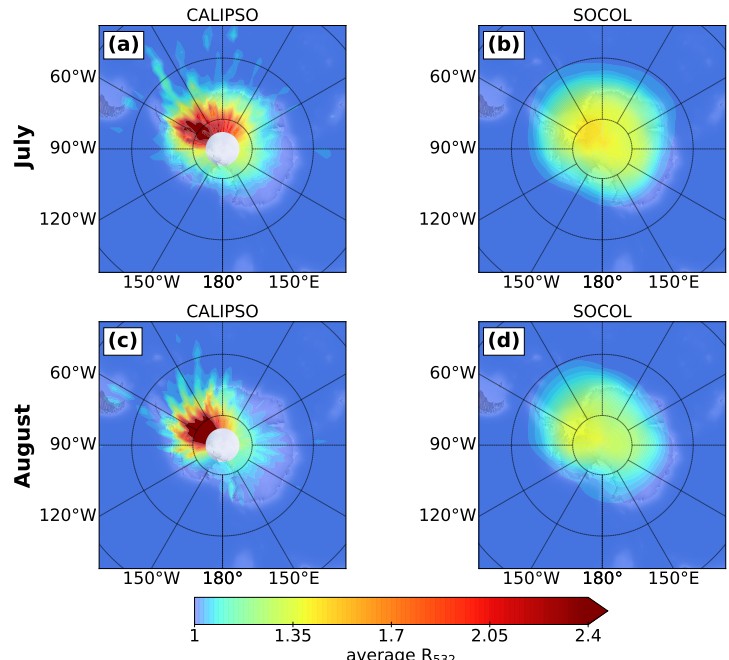

**Figure A4.** As Fig. 4, but for the year 2010.

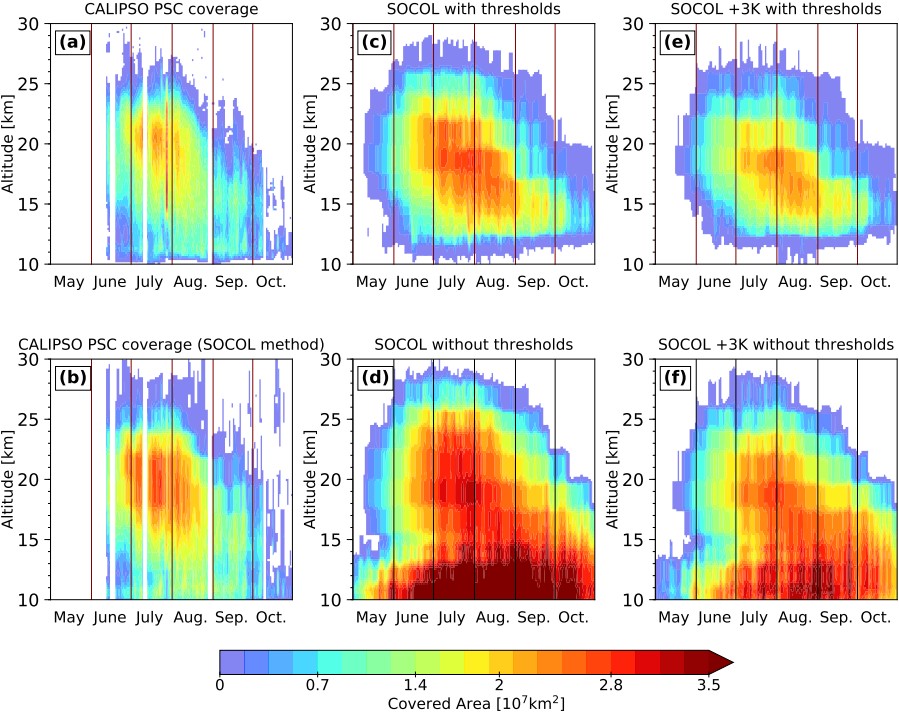

**Figure A5.** As Fig. 5, but for the year 2006.

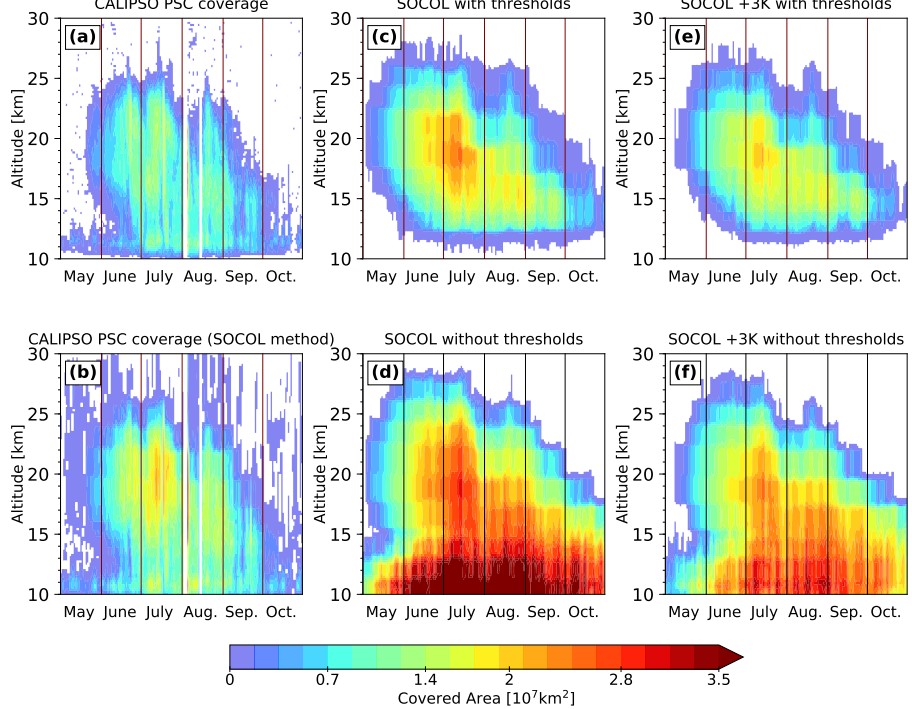

**Figure A6.** As Fig. 5, but for the year 2010.

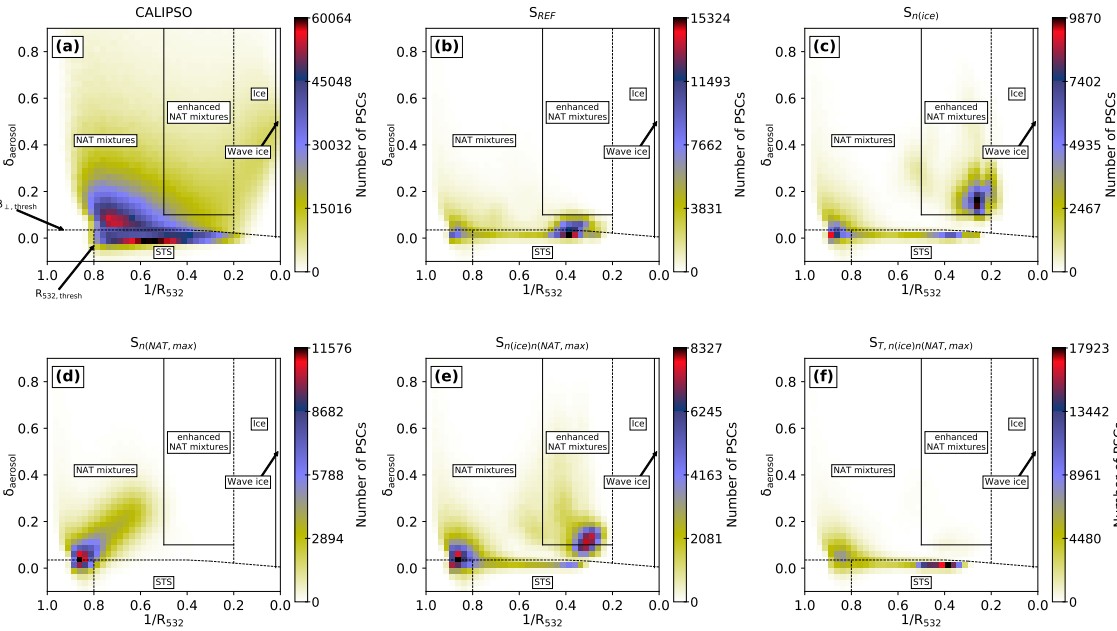

**Figure A7.** Composite 2D-histograms of CALIPSO measurements and SOCOL simulations as in Figs. 1, 6 and A2, but for the year 2006.

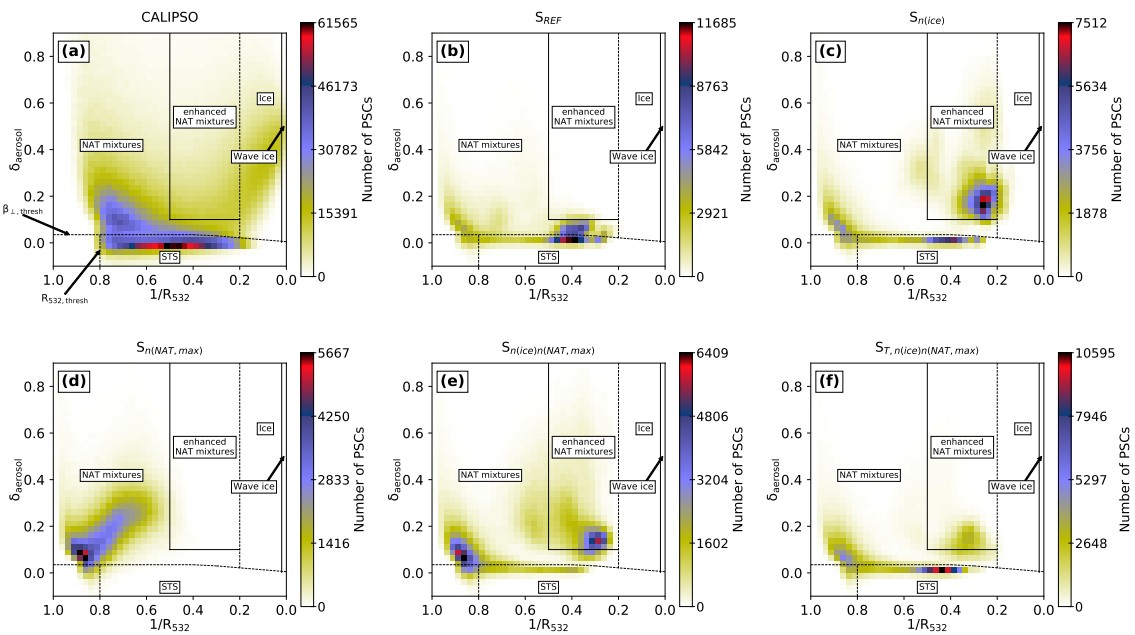

**Figure A8.** As Fig. A7, but for the year 2010.

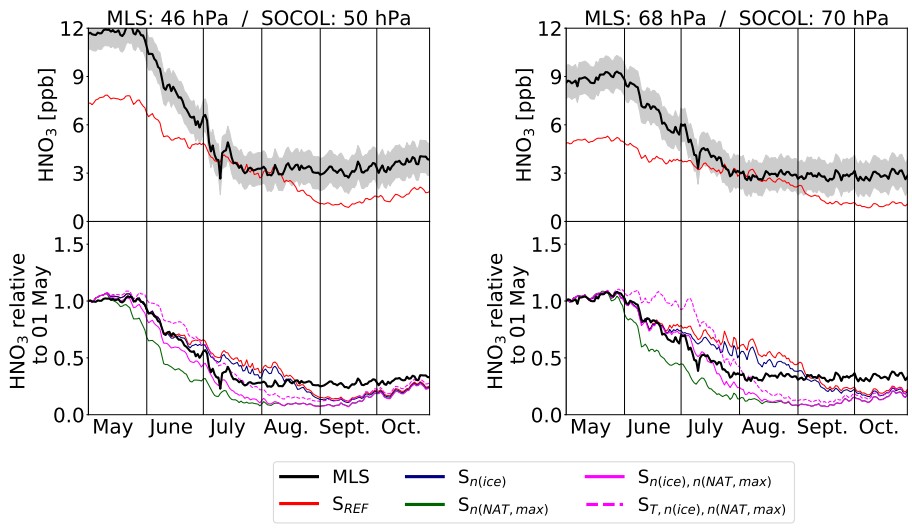

**Figure A9.** As Fig. 7, but for the year 2006.

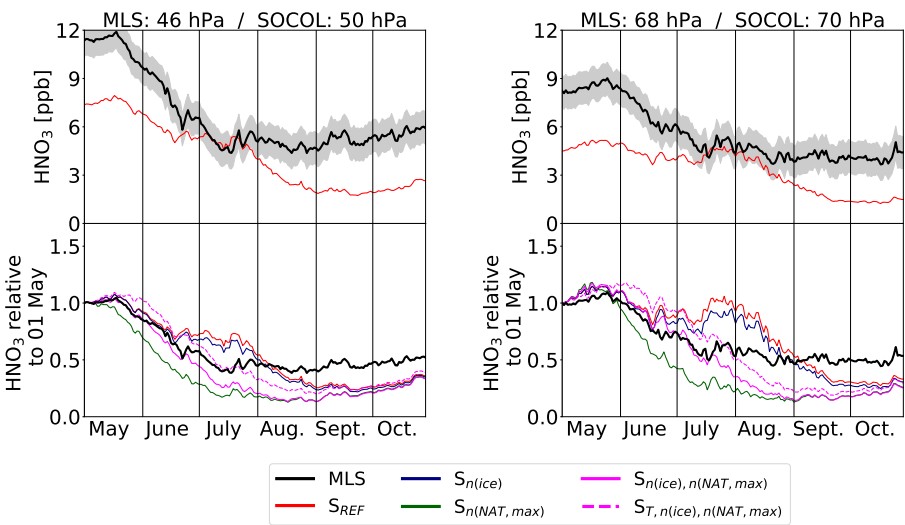

**Figure A10.** As Fig. 7 but for the year 2010.

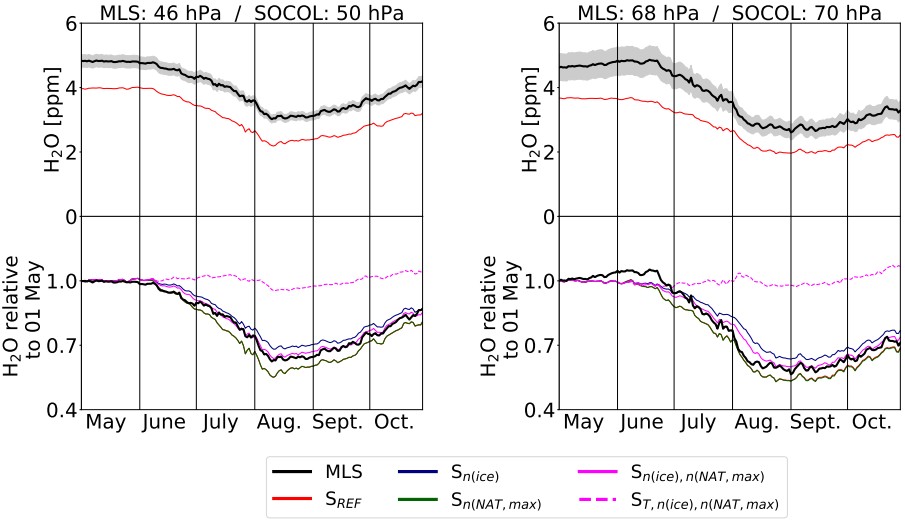

**Figure A11.** As Fig. 8, but for the year 2006.

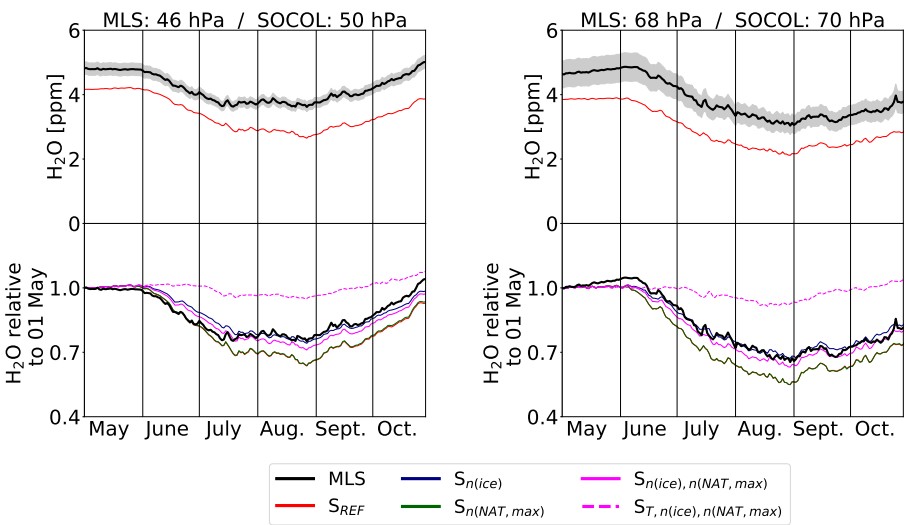

**Figure A12.** As Fig. 8, but for the year 2010.

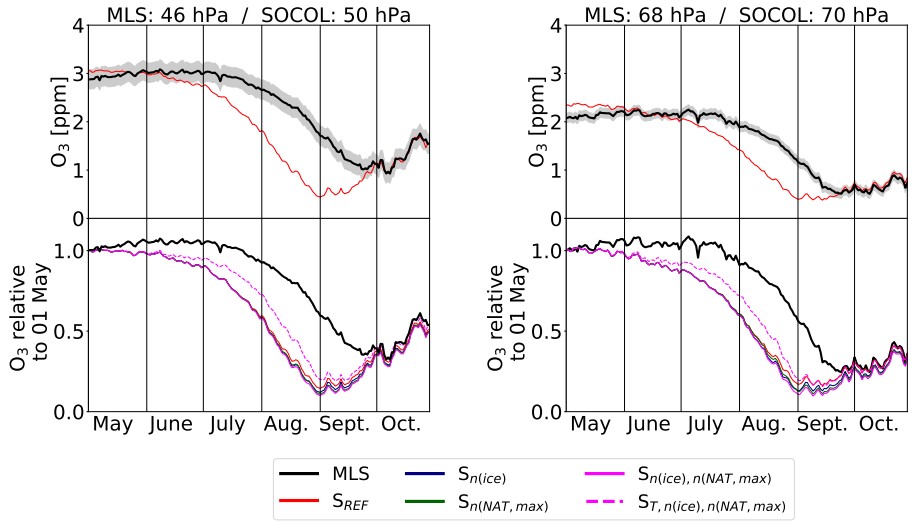

**Figure A13.** As Fig. 9, but for the year 2006.

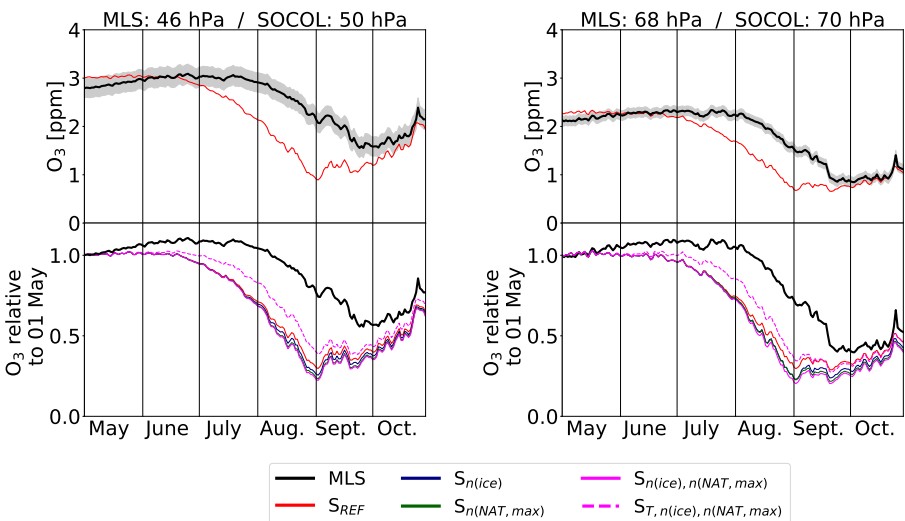

**Figure A14.** As Fig. 9, but for the year 2010.

*Author contributions.* AS and TP initiated the project of evaluating the PSC parametrization in SOCOL. AS conducted all SOCOL simulations. BL provided the Mie and T-matrix scattering code. MS analyzed the model results and wrote the paper. MP provided the CALIOP v2 data. All coauthors helped with the interpretation of the data and contributed to the manuscript.

*Competing interests.* The authors declare that they have no conflict of interest.

*Acknowledgements.* Special thanks go to Beiping Luo for providing the Mie and T-matrix scattering code. We thank Aryeh Feinberg and Franziska Zilker for their technical assistance and support. We also gratefully acknowledge Lamont Poole at NASA Langley Research Center for his help with the CALIPSO figures.

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
