# Peer review of "Evaluation of polar stratospheric clouds in the global chemistry-climate model SOCOLv3.1 by comparison with CALIPSO spaceborne lidar measurements"

_Geoscientific Model Development, 2020_

## Short Comment (SC1) · 29 Jun 2020

Dear authors,

in my role as Executive editor of GMD, I would like to bring to your attention our Editorial version 1.2:

https://www.geosci-model-dev.net/12/2215/2019/

This highlights some requirements of papers published in GMD, which is also available

on the GMD website in the 'Manuscript Types' section:

http://www.geoscientific-model-development.net/submission/manuscript_types.html

In particular, please note that for your paper, the following requirements have not been fully met in the Discussions paper:

- "Code must be published on a persistent public archive with a unique identifier for the exact model version described in the paper or uploaded to the supplement, unless this is impossible for reasons beyond the control of authors. All papers must include a section, at the end of the paper, entitled "Code availability". Here, either instructions for obtaining the code, or the reasons why the code is not available should be clearly stated. It is preferred for the code to be uploaded as a supplement or to be made available at a data repository with an associated DOI (digital object identifier) for the exact model version described in the paper. Alternatively, for established models, there may be an existing means of accessing the code through a particular system. In this case, there must exist a means of permanently accessing the precise model version described in the paper. In some cases, authors may prefer to put models on their own website, or to act as a point of contact for obtaining the code. Given the impermanence of websites and email addresses, this is not encouraged, and authors should consider improving the availability with a more permanent arrangement. Making code available through personal websites or via email contact to the authors is not sufficient. After the paper is accepted the model archive should be updated to include a link to the GMD paper."

So I understand, that the SOCOL code can not be made publicly available due to the restriction of an ECHAM5 license, it is not sufficient to provide a personal email as contact for the SOCOL code. Even though the code can not be made publicly available, it would be possible to provide a permanent landing page (e.g. to a closed project by Zenodo, where than a contact email could be provided.)

The important point is, that a personal email address can change very fast (the person leaves the institute, that institute changes the email address etc.), with a doi-ed landing page it would be possible to update the contact information there, in case the original email is not valid anymore.

Yours,

Astrid Kerkweg

---

## Referee Comment (RC1) · Anonymous Referee #1 · 1 Jul 2020

The authors present an interesting study of comparison of the model outputs of the SOCOL model with observations by the satellite-borne lidar CALIOP. The approach is to test if a CCM without a detailed microphysical model for the formation of PSCs can be used to calculate PSCs in the polar regions. The advantage of such an approach is the reduced time for calculations wrt more sofisticated models including microphysical schemes. To demonstrate the merits and deficits of such an approach the model output is processed to obtain optical parameters which allow PSC classification similar to that used by CALIOP. The authors compare the optical constants measured by CALIOP

with those obtained from the SOCOL model. How are these optical parameters obtained? The authors state "From the simulated SADs and the assumed microphysical parameters, we calculate the number density and/or radius for each particle type.". They also state that the radius of the NAT and STS particles is fixed (5 micron for NAT but we don't know for STS), and that ice has a variable radius, but we don't know how this is obtained. ("The variable radius of ice particles results in a variable _aerosol-value."). Since the conversion of SAD to particle size distribution and number density has an important impact on the results, the authors should dedicate a paragraph on how this is done. Why don't they use a size distribution for all particles, instead of applying observational uncertainties to the results of the Mie calculations? This is of course an artificial way to obtain some scattering of the results but it is not equivalent to using a size distribution. Also by fixing the radius for NAT, the sedimentation velocity is the same for all NAT particles, while for a size distribution the sedimentation velocity would be also a distribution. . .. So to my opinion, the inclusion of a size distribution for all PSC particles would give a more realistic approach and would not make the calculations much more time consuming. I don't understand "but at the end of each chemical time step all condensed HNO3 and H2O evaporates back to the gas phase."

———————————————————————

---

## Referee Comment (RC2) · Anonymous Referee #2 · 6 Jul 2020

Steiner et al. present an evaluation of PSCs simulated with the CCM SOCOL by comparing these to the backscatter observations derived with CALIOP onboard CALIPSO. The comparison is performed for the Antarctic winter 2007 and simulation performance has been tested by using different microphysical properties to optimize the set-up for the PSC scheme.

This is a quite interesting study and could definitely be useful for the modelling community. However, I am a bit disappointed with the outcome of this study. At the

end, with none of the used set-up a really good agreement with observations is found so that not really a recommendation for the modelling community can be given. On top of that the impact on the results for ozone (which was used as a motivator for this study) is not there at all. After major revisions the manuscript may be suitable for publication. Detailed comments for improvement are provided below.

**General comments:**
Why has the Anatarctic winter 2007 been chosen? Is this winter representative for Antarctic winters? Why is only one winter analysed on not several? From the simulations and observations more years should be available. The SOCOL data is modified so that it mimics what CALIPSO is measuring. However, since SOCOL has the much coarser resolution wouldn't it then better to try modify the CALIPSO data so that it rather mimics the SOCOL coarse resolution and therefore what SOCOL is simulating?

The results should be put in the context of results derived from other studies for discussing and understanding the differences between simulation and observation (e.g. Khosrawi et al. 2018 for the Arctic and the papers by Orr et al. 2015 for the Antarctic). Also the efforts done by the WACCM community to improve the PSC scheme could be helpful for the discussion (Wegner et al. 2013; Brakebusch et al. 2013).

The underestimation of denitrification seems to be a general problem in GCMs. This underestimation was also found in Khosrawi et al. 2017 and 2018 comparing EMAC with MLS and still remained even with a higher resolution. In general you blame to often the coarse resolution, but forgot to consider that also deficiencies in the model physic play a role as the representation of dynamics (e.g. descend) and the interplay of the chemistry.

**Specific comments:**

P1, L2: The process of denitrification (namely sedimentation of PSC particles and thus HNO3 removal from this atmospheric layer) should be quickly explained (as has been done in the introduction).

P1, L18: Which resolution has been used? Add here T42L39.

P2, L31: I think there are even newer references. There is at least the paper by Nakajima et al. (2016).

P2, L40: Is there really no newer version of the PSC scheme? Please check.

P4, L 99: With satellite observations from which satellite? MLS? Please add this information.

P4, L108: Why is the hydrolysis of N2O5 important? This should be explained.

P4, L113: PSC types……... -> this is a repetition. This has already mentioned in the previous paragraph.

P4, L128: "….but at the end of each chemical time step all condensed HNO3 and H2O evaporates back to the gas phase". What do you mean with that? This is not realistic at all.

P5, L134: Using 39 vertical levels is really coarse and what is the motivation for doing this. Several studies show that much better results are derived with a higher resolution and computer resources nowadays allow doing such simulations. Especially since you only consider one winter you could have easily done the simulations with a much better resolution. Especially, the coarse vertical resolution is a drawback. Why

have you not used 90 leveles? Using 90 levels significantly improves the results in the stratosphere.

P9, L225: Much is attributed to the coarse resolution. However, why has such a coarse resolution been used? Why has not one of the used set-up been used for a simulation with a higher resolution to check what impact this would have?

P9, 234: PSC formation depends strongly on temperature. How well is temperature simulated in SOCOL?

P12, Table 1: How are these values motivated? Have these been derived from the CALIPSO measurements or are these based on what is used in the literature (based on other observations or other experience with model simulations)?

P12, L274: Add references. At least there is a publication by Grooss et al. where a certain value for the ice number density has been used.

P13, L312: I do not really understand how this is done. How do you account for the heterogeneity of the MLS data by using area-weighted concentrations for SOCOL? How does that mimic the MLS measurements? Why not using the averaging kernels of MLS or just using the SOCOL output at the locations of the MLS measurements (thus along the satellite orbits)?

P16, L354 and 358: As stated in my general comments. The differences in agreement are partly caused by the coarse resolution. There are many other factors playing a role as well.

P17, L369: But what is then the usefulness of this study? What would you recommend the modelers to do to improve their simulation results?

P17, L374: This is nothing new. This has also presented in other studies (e.g. Orr et al., Wegner et al., Brakebusch et al., Khosrawi et al .)

P1, L3: concentrations -> occurrences

P2, L41: PSCs are observed -> I that context I would rather write PSCs can be observed

P3, L84: The acronym MIPAS has not been introduced.

P5, L140: 01 -> 1 (that should be changed in the text throughout the manuscript)

P6, L183-184: repetition of vertical resolution

P8, L213: 01 -> 1

P11, L250: 77.4-90° -> 77-90°

Figure A1 caption: 1st -> 1

Figure 2-9, A1: The resolution of the figures is not good enough. On my printed version the plot frames are missing in several occasions.

Figure 4: Swap the upper panels with the lower ones, so that the order is July, August. Why do you use SREF as figure title? Why not using "SOCOL" as figure title?

Figure 5: Here I would suggest to change the figure titles as follows: "CALIPSO", "SOCOL with thresholds", "SOCOL without thresholds".

Figures 7 and 8: Observations and model simulations are difficult to distinguish. I would suggest to use a thicker line for the observations and maybe a different color.

**References:**

Brakebusch, M., Randall, C.ÂăE., Kinnison, D.ÂăE., Tilmes, S., Santee, M.ÂăL., and Manney, G.ÂăL.: Evaluation of Whole Atmosphere Community Climate Model simulations of ozone during Arctic winter 2004–2005, J.ÂăGeophys. Res., 118, 2673–2688, https://doi.org/10.1002/jgrd.50226, 2013.

Khosrawi, F., Kirner, O., Sinnhuber, B.-M., Johansson, S., Höpfner, M., Santee, M. L., Froidevaux, L., Ungermann, J., Ruhnke, R., Woiwode, W., Oelhaf, H., and Braesicke, P.: Denitrification, dehydration and ozone loss during the 2015/2016 Arctic winter, Atmos. Chem. Phys., 17, 12893–12910, https://doi.org/10.5194/acp-17-12893-2017, 2017.

Khosrawi, F., Kirner, O., Stiller, G., Höpfner, M., Santee, M. L., Kellmann, S., and Braesicke, P.: Comparison of ECHAM5/MESSy Atmospheric Chemistry (EMAC) simulations of the Arctic winter 2009/2010 and 2010/2011 with Envisat/MIPAS and Aura/MLS observations, Atmos. Chem. Phys., 18, 8873–8892, https://doi.org/10.5194/acp-18-8873-2018, 2018.

Nakajima, H., Wohltmann, I., Wegner, T., Takeda, M., Pitts, M. C., Poole, L. R., Lehmann, R., Santee, M. L., and Rex, M.: Polar stratospheric cloud evolution and chlorine activation measured by CALIPSO and MLS, and modeled by ATLAS, Atmos. Chem. Phys., 16, 3311–3325, https://doi.org/10.5194/acp-16-3311-2016, 2016.

Orr, A., Hosking, J. S., Hoffmann, L., Keeble, J., Dean, S. M., Roscoe, H. K., Abraham, N. L., Vosper, S., and Braesicke, P.: Inclusion of mountain-wave-induced

cooling for the formation of PSCs over the Antarctic Peninsula in a chemistry–climate model, Atmos. Chem. Phys., 15, 1071–1086, https://doi.org/10.5194/acp-15-1071-2015, 2015.

Wegner, T., Kinnison, D.ĂăE., Garcia, R.ĂăR., and Solomon, S.: Simulation of polar stratospheric clouds in the specified dynamics version of the whole atmosphere community climate model, J.ĂăGeophys. Res., 118, 4991–5002, https://doi.org/10.1002/jgrd.50415, 2013.

---

## Short Comment (SC2) · 6 Jul 2020

Hello,

It is really exciting to see someone working on PSCs as I do. Sorry to put extra work on you but I hope my comments can help you prepare your paper better.

I mainly have two major comments on the manuscript.

1. I'm not convinced by your reasonable denitrification and ozone simulation right now

(abstract Line 20). I think adding error bars (MLS accuracy) to the observation on your Figures 7, 8, and 9 may help to see whether the modeled HNO3, H2O, and O3 are reasonable. Right now, the onset of O3 depletion seems much earlier than the observation. This is important since the onset date of O3 depletion is one of the important indicators for ozone recovery (Solomon et al., 2016). Is your early O3 depletion caused by your early PSC formation that provides more SAD (Line 259)? I think your cold bias in the model may also contribute to both early PSC formation and early O3 depletion. If your model is not consistent with the O3 and other species after you add the error bars, you may want to emphasize your conclusion on "The change of NAT scheme has minimum impact on O3 depletion." And this conclusion is supported by previous studies like Tabazadeh et al.2000 which find the denitrification impact on Arctic ozone depletion but not much on the Antarctic one.

2. You explain your mismatching of PSCs to CALIPSO is due to the wave PSCs (e.g. Line 240, Line 245). I think the explanation is not enough. Why does the mountain wave cause higher R532? It is not just because of the wave-ice cloud, since wave ice clouds are a very small portion of PSCs. The large R532 is likely to be enhanced-mix or ice clouds in CALIPSO observation. It is probably because you exclude NAT particles with higher number densities. These NAT particles are generated from ice or wave-ice cloud downwind the Antarctic Peninsula. Many observations saw or retrieved small NAT particles ($\sim$2 um) with large number densities, as well as the model simulations (see Zhu et al., 2017, and references therein, note that this is not the same paper you cited in this manuscript). Zhu, Y., Toon, O. B., Lambert, A., Kinnison, D. E., Bardeen, C., & Pitts, M. C. (2017). Development of a polar stratospheric cloud model within the Community Earth System Model: Assessment of 2010 Antarctic winter. Journal of Geophysical Research: Atmospheres, 122, 10,418– 10,438. https://doi.org/10.1002/2017JD027003 Line 295, I suggest you ran a test case with increased Sn(NAT, max) but decrease the NAT size.

And Some small comments, mainly need more citations:

1. In the abstract, you mentioned meteoric dust as a possibility for PSC formation, but you haven't talked about it at all in the main content. Maybe put something in the discussion session.

2. Line 64, please cite Wegners et al., 2012 for PSC parameterization in WACCM

3. Line 65, I think Bardeen 2013 is not relevant.

4. Line 65 and 78-82, This is not the newest publication from Zhu et al. Please cite: Zhu, Y., Toon, O. B., Lambert, A., Kinnison, D. E., Bardeen, C., & Pitts, M. C. (2017). Development of a polar stratospheric cloud model within the Community Earth System Model: Assessment of 2010 Antarctic winter. Journal of Geophysical Research: Atmospheres, 122, 10,418– 10,438. https://doi.org/10.1002/2017JD027003 In this paper, we improved the model with ice to NAT nucleation. And the model is able to capture the small NAT particle features and compare pretty well with CALIPSO backscatter.

5. Line 94, the equilibrium scheme is only for STS, but not for NAT and ice. Please rephrase here.

6. Line 119, could you provide a citation for "observational evidence"?

7. Line 140, instead of "average year", you may say that 2007 is a typical Antarctic year with a steady vortex and observed PSCs from May to September. It is a year without the impact of volcanic eruptions. I think it would ask one question from another referee.

8. Line 194, please list citations for these refractive index numbers.

9. Line 213, I think you mean "Figure 3d"

10. Figure 3a-c: these three figures have very different color bars. I cannot tell if you have a good comparison with CALIPSO or not. Instead of R532, you may use 1/R532 so you don't have to compensate your color bar due to the wave-ice cloud. It's up to you.

11. Line 224 and 283, It is not due to "orography". It is due to the "lack of orographic

gravity representation in the model".

12. Line 257, Is your CALIPSO figure identical with Pitts 2018? if so, Pitts 2018 says 77.8 rather than 77.4.

12. Line 265, "contributes to the larger PSC area and longer period".

13. Line 267-269, This sentence is not logical enough. Even you filtered it when you comparing to CALIPSO observation, you still count them into the SAD in your model, right? You may want to say "these STS clouds contribute to negligible SAD to the ozone chemistry in the model" if this statement is true.

14. Line 292, need a citation here for ice PSCs are less important for stratospheric ozone chemistry.

15. Line 320, "underestimates the HNO3 compared to MLS"

————————————————

---

## Author Comment (AC1) · 6 Nov 2020

Reply to interactive comment by Astrid Kerkweg

Dear Dr. Kerkweg,

we fully understand the point that a permanent landing page allows for updating the contact information for the model source code, while this is not possible in a published paper.

[Figure]

Therefore, we have included the contact information for the SOCOLv3.1 model code to the zenodo repository (http://doi.org/10.5281/zenodo.4094663) and changed the code availability section accordingly. It now reads:

"Since the full SOCOLv3.1 code is based on ECHAM5, users must first sign the ECHAM5 license agreement before accessing the SOCOLv3.1 code (http://www.mpimet.mpg.de/en/science/models/license/, last access: 2020). Then the SOCOLv3.1 code is freely available. The contact information for the full SOCOLv3.1 code as well as the source code of the PSC module and the Mie and T-matrix scattering code are available at http://doi.org/10.5281/zenodo.4094663. . . ."

We hope this fulfils the requirements.

Please note that in addition we uploaded two zip-archives including coefficients for the T-matrix calculations that were missing in the initial repository. Therefore, the doi has changed.

Best regards, Michael Steiner

──────────────────────────────

---

## Author Comment (AC2) · 6 Nov 2020

**Author response to comments of Referee 1**

We thank the referee for taking the time to read the manuscript and for the helpful feedback. Although we think that many most of the points raised are already described in the manuscript, it became clear that some clarification is necessary. In response to Referee 1, along with other revisions in response to Referee 2, we extended the description of our PSC scheme (Sect. 2.1) and of our evaluation approach (Sect. 2.4).

[Figure]

We hope that our answers and revisions help to clarify the description. We present our responses below, with reviewer comments in blue and author responses in black.

The authors present an interesting study of comparison of the model outputs of the SOCOL model with observations by the satellite-borne lidar CALIOP. The approach is to test if a CCM without a detailed microphysical model for the formation of PSCs can be used to calculate PSCs in the polar regions. The advantage of such an approach is the reduced time for calculations wrt more sophisticated models including microphysical schemes. To demonstrate the merits and deficits of such an approach the model output is processed to obtain optical parameters which allow PSC classification similar to that used by CALIOP. The authors compare the optical constants measured by CALIOP with those obtained from the SOCOL model. How are these optical parameters obtained? The authors state "From the simulated SADs and the assumed microphysical parameters, we calculate the number density and/or radius for each particle type.".

The idea of our study is to evaluate the PSCs simulated by SOCOLv3.1 with the help of backscatter measurements by CALIOP onboard the CALIPSO satellite. For that purpose, we converted the simulated PSCs quantities, namely the SAD of STS, NAT and ice, into a size distribution and calculated the optical signal CALIPSO would measure. This is described in Sect. 2.4.

The general procedure is the following: In SOCOL, NAT and water ice are calculated as soon as the partial pressures of $HNO_3$ and water vapor, respectively, exceed supersaturation. From the excess $HNO_3$ and $H_2O$, the surface area density of NAT and water ice is calculated. Herein we assume for NAT a fixed radius and a maximum number density. The latter assumption prevents that all excess $HNO_3$ goes into NAT particles at the expense of STS formation. This accounts for the fact that NAT and STS clouds are mostly observed simultaneously (e.g. Pitts et al., 2011). Conversely, for water ice we assume a fixed number density and calculate the radius from the total ice volume.

[Figure]

The rationale behind the different treatment of NAT and water ice in the model is the following: For water ice, the time to reach equilibrium between the gas- and particulate phase is very short. That means that, once ice has formed, the ice number density stays quite constant and further cooling leads rather to particle growth than to additional nucleation. In the case of NAT, however, the equilibrium between the gas and particulate phase is hardly reached, as shown by observations (Fahey et al., 2001), and additional particles can nucleate upon further cooling. Therefore, we do not fix the NAT number density, but the radius, which has been optimized to match observed sedimentation/denitrification. We are aware that this bulk parameterization is a simplification of the real world, but it helps keeping computational effort low. To various extents, this is done in most CCMs. Thus, the strongest point of the present analysis is the comparison with the state-of-the-art satellite data.

The basis for STS droplets are binary $H_2O$-$H_2SO_4$ aerosol particles, whose distribution is prescribed from a monthly mean observational data record, mainly based on SAGE-observations. The data record provides SAD, volume density, mean radius and $H_2SO_4$ mass of the binary aerosol. STS droplets form in the model when gas-phase $HNO_3$ and $H_2O$ is taken up by the binary aerosols, following the expression by Carslaw et al. (1995). The uptake of $HNO_3$ and $H_2O$ leads to a change in aerosol mass, from which a growth factor of the binary aerosol and therefore the radius of the ternary aerosol can be calculated.

For all three PSC type we outputted the individual surface area density for each model grid point. For the present study we used a 12 hourly output frequency.

To mimic the satellite measurements we proceeded as follows: From the outputted SADs of the three PSC types and the above mentioned assumptions on NAT radius and water ice number density as well as with the STS-radius resulting from the above mentioned growth-factor for ternary aerosols, we calculate the missing parameter, i.e. the number density or radius. These quantities are then used in Mie and T-matrix scattering codes (Mishchenko et al., 1996) to compute optical parameters of the simulated PSCs,

i.e. R532, $\delta_{aerosol}$ and $\beta\perp$. As shapes we assume ellipsoids with an aspect ratio of 0.9 (diameter-to-length ratio) for NAT and ice. STS are liquid particles and therefore assumed to be spherical with a depolarization ratio $\delta_{aerosol}$ = 0.

They also state that the radius of the NAT and STS particles is fixed (5 micron for NAT but we don't know for STS), and that ice has a variable radius, but we don't know how this is obtained. ("The variable radius of ice particles results in a variable $\delta_{aerosol}$ value."). Since the conversion of SAD to particle size distribution and number density has an important impact on the results, the authors should dedicate a paragraph on how this is done.

As mentioned above only the radius of the NAT particles is fixed, not the radius of STS-particles. We noticed that our formulation in the manuscript was indeed very misleading and rephrased the sentence "STS and NAT, due to their spherical shape and fixed radius, appear at constant $\delta_{aerosol}$ -values of 0 and 0.167, respectively." to "STS (due to their spherical shape) and NAT (due to the assumed fixed radius) appear at constant $\delta_{aerosol}$ -values of 0 and 0.167, respectively."

Why don't they use a size distribution for all particles, instead of applying observational uncertainties to the results of the Mie calculations? This is of course an artificial way to obtain some scattering of the results but it is not equivalent to using a size distribution. Also by fixing the radius for NAT, the sedimentation velocity is the same for all NAT particles, while for a size distribution the sedimentation velocity would be also a distribution. . .. So to my opinion, the inclusion of a size distribution for all PSC particles would give a more realistic approach and would not make the calculations much more time consuming.

Thanks for the question. The reason for adding the noise level of the satellite data to the results of the Mie and T-matrix calculations is that the satellite observations include significant uncertainty, i.e. a CALIPSO measurement of even a monodisperse PSC distribution would show a lot of scatter. To mimic these observational uncertainties we

added the instrumental noise to our results.

Concerning the distribution of sedimentation velocities, we agree that there might be better approaches to describe the size distribution of PSC particles. However, the purpose of this paper is not to come up with a microphysically fully consistent PSC size distribution, but to evaluate and optimize the existing parameterization. Instead of fixing the NAT radius, other models fix the NAT number density (e.g. Wegner et al., 2013, Nakajima et al., 2016), which results in varying NAT radius and sedimentation velocities. However, the value for NAT number density is the model dependent and acts as tuning parameter. In reality, the actual value for NAT number density is far from being constant, because the active sites for NAT nucleation themselves show a wide distribution of efficacies (Hoyle et al., 2013). Therefore, we follow a different approach and choose a NAT radius to reasonably simulate the observed sedimentation/denitrification features.

I don't understand "but at the end of each chemical time step all condensed HNO3 and H2O evaporates back to the gas phase."

This means that the NAT and water ice particles are not themselves prognostic variables and are not explicitly transported by the model's advection scheme. This is a common approach in CCMs. At the end of the chemistry routine, the condensed HNO3 and H2O is returned to the gas phase and the transport occurs via the gas phase. At the next call of the chemistry scheme, NAT is freshly formed if the partial pressure of HNO3 exceeds supersaturation, and the particles are re-established within this equilibrium scheme. The same holds for water ice and the partial pressure of H2O. This procedure prevents numerical diffusion of within and between model grid cells of HNO3 and H2O, as PSC clouds are regionally limited and show strong gradients. We will rephrase the sentence for clarification.

**References:**

Carslaw, K. S., Luo, B. P., and Peter, T.: An analytic expression for the composition of

aqueous HNO3–H2SO4 stratospheric aerosols including gas-phase removal of HNO3, Geophys. Res.Lett., 22, 1877–1880, 1995.

Fahey, D. W., Gao, R. S., Carslaw, K. S., Kettleborough, J., Popp, P. J., Northway, M. J., Holecek, J. C., Ciciora, S. C., McLaugh-lin, R. J., Thompson, T. L., Winkler, R. H., Baumgardner, D. G., Gandrud, B., Wennberg, P. O., Dhaniyala, S., McKinney, K., Peter, T., Salawitch, R. J., Bui, T. P., Elkins, J. W., Webster, C. R., Atlas, E. L., Jost, H., Wilson, J. C., Herman, R. L., Kleinböhl, A., and von König, M.: The detection of large HNO3-containing particles in the winter arctic stratosphere, Science, 291, 1026–1031, https://doi.org/10.1126/science.1057265, 2001.

Hoyle, C. R., Engel, I., Luo, B. P., Pitts, M. C., Poole, L. R., Grooß, J.-U., and Peter, T.: Heterogeneous formation of polar stratospheric clouds- Part 1: Nucleation of nitric acid trihydrate (NAT), Atmos. Chem. Phys., 13, 9577–9595, https://doi.org/10.5194/acp-13-9577-2013, 2013.

Nakajima, H., Wohltmann, I., Wegner, T., Takeda, M., Pitts, M. C., Poole, L. R., Lehmann, R., Santee, M. L., and Rex, M.: Polar stratospheric cloud evolution and chlorine activation measured by CALIPSO and MLS, and modeled by ATLAS, Atmos. Chem. Phys., 16, 3311–3325, https://doi.org/10.5194/acp-16-3311-2016, 2016.

Pitts, M. C., Poole, L. R., Dörnbrack, A., and Thomason, L.W.: The 2009–2010 Arctic polar stratospheric cloud season: a CALIPSO perspective, Atmos. Chem. Phys., 11, 2161–2177, https://doi.org/10.5194/acp-11-2161-2011, 2011.

Wegner, T., D. E. Kinnison, R. R. Garcia, and S. Solomon (2013), Simulation of polar stratospheric clouds in the specified dynamics version of the whole atmosphere community climate model, J. Geophys. Res. Atmos., 118, 4991–5002, doi:10.1002/jgrd.50415.

---

## Author Comment (AC3) · 6 Nov 2020

Dear Yunqian Zhu,

thank you very much for your helpful feedback. We appreciate your suggestions, which helped to improve our manuscript a lot. We present our point-by-point reply below, with your comments in blue and our responses in black.

Best regards, Michael Steiner

[Figure]

**Major comments:**

I'm not convinced by your reasonable denitrification and ozone simulation right now (abstract Line 20). I think adding error bars (MLS accuracy) to the observation on your Figures 7, 8, and 9 may help to see whether the modeled HNO3, H2O, and O3 are reasonable. Right now, the onset of O3 depletion seems much earlier than the observation. This is important since the onset date of O3 depletion is one of the important indicators for ozone recovery (Solomon et al., 2016). Is your early O3 depletion caused by your early PSC formation that provides more SAD (Line 259)? I think your cold bias in the model may also contribute to both early PSC formation and early O3 depletion. If your model is not consistent with the O3 and other species after you add the error bars, you may want to emphasize your conclusion on "The change of NAT scheme has minimum impact on O3 depletion." And this conclusion is supported by previous studies like Tabazadeh et al.2000 which find the denitrification impact on Arctic ozone depletion but not much on the Antarctic one.

We added the MLS error bars to the Figures 7-9. While at the beginning of winter, O3 agrees well with observations, the difference between simulations and observations for HNO3 and H2O are large and also larger than the MLS uncertainty (upper panel). However, when looking at the relative evolution of the species (lower panel of the figures), the amplitude of denitrification and dehydration is in good agreement with observations, depending on the simulation. This is different for O3, where depletion in the model occurs too early and too strong, as you correctly stated. To test the hypothesis that the early onset of O3 depletion in the model is caused by early PSC formation, we ran sensitivity simulations with the temperature in the PSC routine constantly increased by 3K, which is roughly the temperature bias in the lower stratosphere. We indeed see a later onset of the PSC-season and a reduced PSC area, both of which is in better agreement with CALIOP-observations. As a consequence, the onset of O3 depletion is indeed delayed by slightly less than 1 month. However, this is still earlier than observed by MLS. Towards the end of the winter, the differences between the simulations vanish. The relatively small impact of the PSC scheme on the underestimation of O3 likely is a result of an underestimated downward transport inside the polar vortex. This is also discussed in the answers to the referee comment 2. We added this point to our discussion and included the fact, that modifications in the NAT scheme have minimal impact on O3, to our conclusions.

You explain your mismatching of PSCs to CALIPSO is due to the wave PSCs (e.g. Line 240, Line 245). I think the explanation is not enough. Why does the mountain wave cause higher R532? It is not just because of the wave-ice cloud, since wave ice clouds are a very small portion of PSCs. The large R532 is likely to be enhanced-mix or ice clouds in CALIPSO observation. It is probably because you exclude NAT particles with higher number densities. These NAT particles are generated from ice or wave-ice cloud downwind the Antarctic Peninsula. Many observations saw or retrieved small NAT particles (∼2 um) with large number densities, as well as the model simulations (see Zhu et al., 2017, and references therein, note that this is not the same paper you cited in this manuscript). Zhu, Y., Toon, O. B., Lambert, A., Kinnison, D. E., Bardeen, C., Pitts, M. C. (2017). Development of a polar stratospheric cloud model within the Community Earth System Model: Assessment of 2010 Antarctic winter. Journal of Geophysical Research: Atmospheres, 122, 10,418– 10,438. https://doi.org/10.1002/2017JD027003 Line 295, I suggest you ran a test case with increased Sn(NAT, max) but decrease the NAT size.

Thank you very much for pointing this out. We agree with you that the upper limit for NAT number densities may contribute to the underestimation of the large R532 observations, and we added this point to the discussion of Fig. 4 and Fig. 6. However, we did not perform a further simulation with enhanced N(NAT,max) and smaller r(NAT). Even with such modified NAT parameters we would need some representation of the mountain waves in the model to reproduce these peaks. The observed clouds with enhanced-mix and ice downwind of the Peninsula form on ice which forms due to the mountain waves, and since mountain waves are almost not present in the simulation

we would not see a peak downwind of the Peninsula. We would just shift the R532-values of all grid-boxes containing NAT particles. Furthermore, enhancing N(NAT,max) would put the reasonable agreement with the thin NAT-STS mixtures at stake.

**Minor comments:**

In the abstract, you mentioned meteoric dust as a possibility for PSC formation, but you haven't talked about it at all in the main content. Maybe put something in the discussion session.

We removed the meteoric dust from the abstract. Meteoric dust is not included in our model, and we think that a further discussion is beyond the scope of the paper.

Line 64, please cite Wegners et al., 2012 for PSC parameterization in WACCM

Reference added.

Line 65, I think Bardeen 2013 is not relevant.

We removed the citation.

Line 65 and 78-82, This is not the newest publication from Zhu et al. Please cite: Zhu, Y., Toon, O. B., Lambert, A., Kinnison, D. E., Bardeen, C., Pitts, M. C. (2017). Development of a polar stratospheric cloud model within the Community Earth System Model: Assessment of 2010 Antarctic winter. Journal of Geophysical Research: Atmospheres, 122, 10,418– 10,438, https://doi.org/10.1002/2017JD027003. In this paper, we improved the model with ice to NAT nucleation. And the model is able to capture the small NAT particle features and compare pretty well with CALIPSO backscatter.

Thank you for the hint. We updated the reference and extended the description of the results for WACCM/CARMA.

Line 94, the equilibrium scheme is only for STS, but not for NAT and ice. Please rephrase here.

This is not quite correct. SOCOL does indeed assume equilibrium for STS, NAT and ice. Only if the NAT number density exceeds a certain threshold value, NAT deviates from equilibrium conditions. Since the details of the PSC parameterizations are explained in Sect. 2.1, we decided to leave the respective sentence as is.

Line 119, could you provide a citation for "observational evidence"?

Observational evidence is for example given by the CALIPSO measurements, which show that NAT and STS occur at the same time (Pitts et al., 2011), which would not be possible without HNO3 supersaturation with respect to NAT. Further evidence is shown in the in situ study by Fahey et al. (2001): Fig. 5 D shows that the HNO3 supersaturation ratio with respect to NAT (Snat) was around 15 at the time the NAT particles where sampled (time = 0).

Line 140, instead of "average year", you may say that 2007 is a typical Antarctic year with a steady vortex and observed PSCs from May to September. It is a year without the impact of volcanic eruptions. I think it would ask one question from another referee.

Thanks for the suggestion. We included this statement. Furthermore, we extended our analysis by the years 2006 and 2010, which represent years with above- and below-average PSC occurrence, but also without volcanic influence.

Line 194, please list citations for these refractive index numbers.

1.31 is simply the refractive index for water ice. For NAT, we added the following reference:

Middlebrook, A. M., Berland, B. S., George, S. M., Tolbert, M. A., and Toon, O. B.: Real refractive-indexes of infrared-characterized nitric-acid ice films – implications for optical measurements of polar stratospheric clouds, J. Geophys. Res., 99, 25655–25666, doi:10.1029/94JD02391, 1994.

Line 213, I think you mean "Figure 3d"

Corrected.

Figure 3a-c: these three figures have very different color bars. I cannot tell if you have a good comparison with CALIPSO or not. Instead of R532, you may use 1/R532 so you don't have to compensate your color bar due to the wave-ice cloud. It's up to you.

We adjusted the figure and use now 1/R532 as suggested.

Line 224 and 283, It is not due to "orography". It is due to the "lack of orographic gravity representation in the model".

Corrected.

Line 257, Is your CALIPSO figure identical with Pitts 2018? if so, Pitts 2018 says 77.8 rather than 77.4.

Thanks for spotting this mistake, we fixed it.

Line 265, "contributes to the larger PSC area and longer period".

We adjusted the text accordingly.

Line 267-269, This sentence is not logical enough. Even you filtered it when you comparing to CALIPSO observation, you still count them into the SAD in your model, right? You may want to say "these STS clouds contribute to negligible SAD to the ozone chemistry in the model" if this statement is true.

You are absolutely right, the SAD of all PSC particles is counted in the model. With this sentence we wanted to state that the fact, that these large-scale STS clouds are almost entirely filtered out by the thresholds, shows that these clouds must be teneous. We rephrased this sentence and tried to make the statement more clear:

"Those large-scale STS clouds are very tenuous since they are filtered out when applying the conservative PSC detection threshold. The contribution of those STS clouds to SAD is negligible."

Line 292, need a citation here for ice PSCs are less important for stratospheric ozone chemistry

We rephrased this statement and added a citation: "NAT PSCs play a twofold role in stratospheric ozone chemistry: Besides halogen activation on their surfaces, the sedimentation of NAT particles leads to denitrification, which hinders deactivation of reactive halogens and facilitates catalytic ozone loss (Peter, 1997)."

Line 320, "underestimates the $HNO_3$ compared to MLS"

We adjusted the text accordingly.

**References:**

Fahey, D. W., Gao, R. S., Carslaw, K. S., Kettleborough, J., Popp, P. J., Northway, M. J., Holecek, J. C., Ciciora, S. C., McLaugh-lin, R. J., Thompson, T. L., Winkler, R. H., Baumgardner, D. G., Gandrud, B., Wennberg, P. O., Dhaniyala, S., McKinney, K., Peter, T., Salawitch, R. J., Bui, T. P., Elkins, J. W., Webster, C. R., Atlas, E. L., Jost, H., Wilson, J. C., Herman, R. L., Kleinböhl, A., and von König, M.: The detection of large $HNO_3$-containing particles in the winter arctic stratosphere, Science, 291, 1026–1031, https://doi.org/10.1126/science.1057265, 2001.

Peter, T., Microphysics and heterogeneous chemistry of polar stratospheric clouds, Ann. Rev. Phys. Chem., 48, 785–822, 1997.

Pitts, M. C., Poole, L. R., Dörnbrack, A., and Thomason, L.W.: The 2009–2010 Arctic polar stratospheric cloud season: a CALIPSO perspective, Atmos. Chem. Phys., 11, 2161–2177, https://doi.org/10.5194/acp-11-2161-2011, 2011.
* * *

---

## Author Comment (AC4) · 6 Nov 2020

**Reply to the comment of Referee 2**

We would like to thank the referee for taking the time to read our manuscript and for the helpful feedback. We have taken these comments into account and present our responses below, with reviewer comments in blue and author responses in black. The major changes to our manuscript, as suggested by the reviewer, are:

- Additional model simulations and analyses for the Antarctic winters 2006 and

2010, which represent years with above- and below-average PSC occurrence.

- Additional sensitivity simulations to investigate the impact of the model's cold bias on PSC formation: temperature for PSC formation increased by 3K.

- Extended discussion on further influencing factors for PSC formation like model resolution or temperature biases as well as of previous studies.

Steiner et al. present an evaluation of PSCs simulated with the CCM SOCOL by comparing these to the backscatter observations derived with CALIOP onboard CALIPSO. The comparison is performed for the Antarctic winter 2007 and simulation performance has been tested by using different microphysical properties to optimize the set-up for the PSC scheme.

This is a quite interesting study and could definitely be useful for the modelling community. However, I am a bit disappointed with the outcome of this study. At the end, with none of the used set-up a really good agreement with observations is found so that not really a recommendation for the modelling community can be given. On top of that the impact on the results for ozone (which was used as a motivator for this study) is not there at all. After major revisions the manuscript may be suitable for publication. Detailed comments for improvement are provided below.

In this study we performed the first in-depth evaluation of PSC occurrence and composition in the SOCOL model. We agree that the comparison with the satellite data shows that the agreement is not perfect, but we believe that the modifications in the microphysical parameters have indeed resulted in a significantly improved agreement with the CALIPSO observations. The fact that polar ozone showed little response to the modifications in the PSC scheme was also a surprise to us. From the simulations with enhanced temperatures for PSC formation we saw a later onset of the PSC-season and a reduced PSC area, both of which is in better agreement with CALIOP observations. Consequently, the onset of O3 depletion is also delayed by slightly less than

one month, however still earlier than in the MLS observations. This shows that further improvements of other parts of the model are necessary to reduce the disagreement with between modeled and observed ozone.

**General comments:**

Why has the Antarctic winter 2007 been chosen? Is this winter representative for Antarctic winters? Why is only one winter analysed on not several? From the simulations and observations more years should be available.

We have chosen 2007 since is a typical Antarctic year, with a steady vortex and PSCs from May to September. Furthermore, CALIOP data coverage was high and there was no impact of volcanic eruptions. However, it is absolutely true that, based on available observational data and low computational costs, more winters can be analyzed. In response to the reviewer's criticism, we additionally analyzed the years 2006 and 2010 (with above-average and below-average PSC occurrences, respectively). The analysis of these two additional winters showed very similar results as of 2007 in all comparisons (geographic PSC-distribution, areal coverage, histograms, MLS-comparisons). The main results are the same as for 2007 and the analysis of these two additional winters has not lead to different conclusions. For this reason, we still show the results for the year 2007 in the result-section. However, all plots for the years 2006 and 2010 are included in the appendix of the paper.

The SOCOL data is modified so that it mimics what CALIPSO is measuring. However, since SOCOL has the much coarser resolution wouldn't it then better to try modify the CALIPSO data so that it rather mimics the SOCOL coarse resolution and therefore what SOCOL is simulating?

This is an important point. In the analysis of the spatial distribution, where we show the polar stereographic plots, this averaging of the CALIPSO measurements over the SOCOL grid boxes is already applied. Within this revision, we also apply a similar method for the areal coverage calculations. By doing so, we intend to show by how much the

area calculation from the SOCOL grid boxes contributes to a larger area compared to the method applied by Pitts et al. (2018), especially as we already mentioned in the text that this difference may most likely cause some of the overestimation. Our goal is to calculate the area from the measurements and from the simulations as similar as possible. Therefore, we average the measurements over the SOCOL grid boxes over 12 hours (the output frequency of our simulations). Since not all grid boxes are overpassed within 12 hours, we set the PSC area in the remaining grid boxes to the mean PSC area in the overpassed grid boxes along the same latitude. The areal coverage calculated with this approach is larger than calculated by the method applied in Pitts et al. (2018), which is what we expected. We show this additional Subplot in Fig. 5. However, the simulated PSC area is still larger, which is due to the cold bias of the model. The PSC area from a sensitivity run with increased PSC formation temperature is also included in Fig. 5 to show the effect of the temperature bias on PSC area.

The results should be put in the context of results derived from other studies for discussing and understanding the differences between simulation and observation (e.g. Khosrawi et al. 2018 for the Arctic and the papers by Orr et al. 2015 for the Antarctic). Also the efforts done by the WACCM community to improve the PSC scheme could be helpful for the discussion (Wegner et al. 2013; Brakebusch et al. 2013).

We agree that so far our manuscript was mainly focused on the presentation of our own results. We have now extended our discussion section substantially and discuss several of the mentioned papers. However, it should be mentioned that all these models and their PSC scheme are (slightly) different or the studies focused on different winters/hemispheres. So a one-by-one comparison with our study is not always possible.

The underestimation of denitrification seems to be a general problem in GCMs. This underestimation was also found in Khosrawi et al. 2017 and 2018 comparing EMAC with MLS and still remained even with a higher resolution. In general you blame to often the coarse resolution, but forgot to consider that also deficiencies in the model physic play a role as the representation of dynamics (e.g. descend) and the interplay

of the chemistry.

As mentioned above we have extended our discussion section substantially. We now compare our studies with previous papers and discuss the potential impact of various model deficiencies. We fully agree with the reviewer. The studies by Khosrawi et al. provide very important insights also for our analysis, especially since the EMAC model and the SOCOL model are based on the same general circulation model, namely MA-ECHAM5. EMAC was found to suffer from an underestimation of downward transport inside the polar vortex, and Khosrawi et al. (2017) suggested this as likely reason for the underestimated polar vortex O3. We now compare our studies with previous papers and discuss the potential impact of various model deficiencies. See also below.

**Specific comments:**

P1, L2: The process of denitrification (namely sedimentation of PSC particles and thus HNO3 removal from this atmospheric layer) should be quickly explained (as has been done in the introduction).

Done.

P1, L18: Which resolution has been used? Add here T42L39.

Yes, T42L39 has been used. Information has been added.

P2, L31: I think there are even newer references. There is at least the paper by Naka-jima et al. (2016).

We added the reference Nakajima et al. (2016).

P2, L40: Is there really no newer version of the PSC scheme? Please check.

To our knowledge there is no newer version. The paper by Nakajima et al. (2016) explicitly states that their results confirm the possibility of an ice-free nucleation mechanism of NAT involving solid particles as suggested by Hoyle et al. Furthermore, Naka-jima et al. do not take sedimentation into account. However, we will add the citation.

P4, L 99: With satellite observations from which satellite? MLS? Please add this information.

Yes, from MLS observations. Information will be added.

P4, L108: Why is the hydrolysis of N2O5 important? This should be explained.

In general, the heterogeneous hydrolysis of N2O5 is an important and efficient loss process for NOx as it forms HNO3. The respective reaction in the gas phase is comparatively slow. The heterogeneous reaction is important in the troposphere in aerosol particles and cloud droplets, but also in the stratosphere on binary aerosol and PSC particles. The sentence in our manuscript explicitly refers to the N2O5 hydrolysis on tropospheric aerosols, and was mainly added for the sake of completeness. For the present study, it is not of importance. Therefore, we removed the sentence.

P4, L113: PSC types. . .. . ... -> this is a repetition. This has already mentioned in the previous paragraph.

Sentence will be deleted.

P4, L128: ". . ..but at the end of each chemical time step all condensed HNO3 and H2O evaporates back to the gas phase". What do you mean with that? This is not realistic at all.

This means that the NAT and water ice particles are not explicitly transported by the model's advection scheme. This is a common approach in CCMs. At the end of the chemistry routine, the condensed HNO3 and H2O is re-evaporated and the transport occurs via the gas-phase. At the next call of the chemistry scheme, NAT is freshly formed if the partial pressure of HNO3 exceeds supersaturation. The same holds for water ice and the partial pressure of H2O. This procedure goes back to times when tracer transport was computationally expensive, with the goal to keep the number of prognostic tracers small. Furthermore, it prevents numerical diffusion as PSC clouds are regionally limited and show strong gradients. We will rephrase the sentence for

clarification.

Whether L90 leads to improvements compared to L39 clearly depends on the quantity. We do not see large differences in the simulated Brewer-Dobson-Circulation (w*) between L39 and L90. Furthermore, for the present study we used SOCOL in specified dynamics mode, and in SD mode there are no large differences in stratospheric dynamics between L39 and L90. The cold bias in the polar lowermost stratosphere is very similar in both vertical resolutions. Therefore, we do not expect large differences in the simulated PSCs between L39 and L90. This is also supported by the study of Khosrawi et al. (2017), who applied the EMAC model in L90, in T42 and in a much higher horizontal resolution of T106. Both model versions showed very similar differences to observations. This shows that a higher resolution is not necessarily the remedy for all model deficiencies.

Khosrawi, F., Kirner, O., Sinnhuber, B.-M., Johansson, S., Höpfner, M., Santee, M. L., Froidevaux, L., Ungermann, J., Ruhnke, R., Woiwode, W., Oelhaf, H., and Braesicke, P.: Denitrification, dehydration and ozone loss during the 2015/2016 Arctic winter, Atmospheric Chemistry and Physics, 17, 12 893–12 910, https://doi.org/10.5194/acp-17-12893-2017, https://acp.copernicus.org/articles/17/12893/2017/, 2017.

P9, L225: Much is attributed to the coarse resolution. However, why has such a coarse resolution been used? Why has not one of the used set-up been used for a simulation with a higher resolution to check what impact this would have?

See above. Furthermore, a change to higher horizontal resolutions would require a

complete re-tuning of the model, which is out of the question, also because we are currently working on a new model generation, which will apply T63 as default horizontal resolution. However, even with T63 we will not fully resolve mountain waves.

P9, 234: PSC formation depends strongly on temperature. How well is temperature simulated in SOCOL?

As the majority of chemistry-climate models, SOCOL experiences a cold temperature bias in the polar stratosphere. A comparison with ERA-Interim on four different pressure levels is shown in the Figure below (Fig. 1 of this reply). Mostly, the temperature difference is between 2 and 4 K. To address your question and investigate the impact of the cold bias on PSC formation in the model, we ran two further sensitivity analyses during this revision. In both simulations, temperature for PSC formation was increased by 3K, once for the reference simulation and once for the $S_{n(ice),n(NAT,max)}$ simulation. A discussion of the former simulation in terms of areal coverage and of latter simulation regarding the MLS-comparison has been added to the manuscript. The simulation is denoted as $S_{T,n(ice),n(NAT,max)}$. With the increased PSC formation temperature, PSCs occur later and their area clearly decreased, as expected. The area of this new simulation agrees very well with the observed PSC area (with a similar method as for the simulation; see above). Figure 5 has been extended with these new results. The simulations with increased PSC formation temperature further show a later onset of denitrification and ozone depletion, both of which also is expected with PSC occurring later. However, towards the end of winter, the difference in HNO3 and O3 between the simulations with and without increased PSC formation temperature vanish. Further, the new simulation show almost no more dehydration since ice rarely forms with the formation temperature increased by 3K. The Figures 7-9 now include the new simulation. It is important to highlight, that we didn't increase the temperature of the model itself, but just for the PSC formation (i.e. the tropical tropopause temperature and the related dehydration remain unchanged).

P12, Table 1: How are these values motivated? Have these been derived from the

CALIPSO measurements or are these based on what is used in the literature (based on other observations or other experience with model simulations)?

The default setting for the microphysical parameters has been adopted from the previous model version SOCOL 2 (Schraner et al., 2008). The parameter settings for the sensitivity simulations have been defined based on the evaluation of the reference simulation with CALIPSO measurements and a stepwise modification of $n_{ice}$, $n_{NAT,max}$ and $r_{NAT}$. We did many more simulations than presented in the paper, modified one parameter after the other and analyzed the impact of the respective parameter on the model result. It is clear that in reality PSCs are very heterogeneous in space and time, while a model like SOCOL has a coarse resolution, therefore, the "optimal" parameter setting is always a compromise and requires some testing and tuning. Furthermore, the "optimal" parameter setting will most likely depend on the model resolution or might change in future model versions with different dynamics, treatment of binary aerosol etc.

Schraner, et al., Technical Note: Chemistry-climate model SOCOL: version 2.0 with improved transport and chemistry/microphysics schemes, Atmos. Chem. Phys., 8, 5957–5974, https://doi.org/10.5194/acp-8-5957-2008, 2008.

P12, L274: Add references. At least there is a publication by Grooss et al. where a certain value for the ice number density has been used.

For example, Nakajima et al. (2016) also used 0.01 cm-3. Tritscher et al. (2019 ) used 10 cm-3 for homogeneous ice nucleation under mountain wave conditions. For heterogeneous ice nucleation they use a look-up-table (their Fig. 1) as done in Grooß et al. (2014) for NAT nucleation.

Grooß, J.-U., Engel, I., Borrmann, S., Frey, W., Günther, G., Hoyle,C. R., Kivi, R., Luo, B. P., Molleker, S., Peter, T., Pitts, M.C., Schlager, H., Stiller, G., Vömel, H., Walker, K. A., andMüller, R.: Nitric acid trihydrate nucleation and denitrificationin the Arctic stratosphere, Atmos. Chem. Phys., 14, 1055–1073,https://doi.org/10.5194/acp-14-

1055-2014, 2014.

Tritscher, I., Grooß, J.-U., Spang, R., Pitts, M. C., Poole, L. R., Müller, R., and Riese, M.: Lagrangian simulation of ice particles and resulting dehydration in the polar winter stratosphere, Atmos. Chem. Phys., 19, 543–563, https://doi.org/10.5194/acp-19-543-2019, 2019.

P13, L312: I do not really understand how this is done. How do you account for the heterogeneity of the MLS data by using area-weighted concentrations for SOCOL? How does that mimic the MLS measurements? Why not using the averaging kernels of MLS or just using the SOCOL output at the locations of the MLS measurements (thus along the satellite orbits)?

We average the MLS measurements over each SOCOL grid box, so that it makes the measurements comparable with SOCOL. To calculate the "polar mean" concentrations, the averaged MLS-values as well as the SOCOL concentrations are area-weighted to take into account the different areas of the grid boxes. We changed the sentence to: "To account for the spatial heterogeneity of the MLS measurements, we averaged the measurements over the SOCOL grid boxes from which area-weighted polar mean concentrations are calculated."

P16, L354 and 358: As stated in my general comments. The differences in agreement are partly caused by the coarse resolution. There are many other factors playing a role as well.

Agreed, and now considered in our discussion. See also next point.

P17, L369: But what is then the usefulness of this study? What would you recommend the modelers to do to improve their simulation results?

First, the main goal of the study was to evaluate the PSCs in the SOCOL model, which has never been done before to such an extent. As mentioned above, the fact that O3 did not react very much to the PSC modifications was also a surprise to us and

suggests that other processes than heterogeneous chemistry play an important role for O3 in the polar stratosphere during winter/spring. As pointed out by Khosrawi et al. (2017) model deficiencies in downward transport inside the polar vortex are a promising candidate. As EMAC and SOCOL are both based on MA-ECHAM5 as underlying general circulation model, this might also hold for SOCOL. We mention this now in our discussion. Second, all models are different. It is difficult to come up with a general suggestion for all modelers. Each has to be evaluated individually. We make this point when we put our results in context with other studies.

P17, L374: This is nothing new. This has also presented in other studies (e.g. Orr et al., Wegner et al., Brakebusch et al., Khosrawi et al .)

Agreed. We rephrased this sentence, pointing out that such simplified PSC schemes are widely used in the CCM community, however, with a wide range of assumptions concerning the microphysical parameters.

P1, L3: concentrations -> occurrences

Here we refer to polar ozone, not PSCs, so we think "concentrations" is correct.

P2, L41: PSCs are observed -> I that context I would rather write PSCs can be observed

Sentence has been changed.

P3, L84: The acronym MIPAS has not been introduced.

Acronym is now introduced.

P5, L140: 01 -> 1 (that should be changed in the text throughout the manuscript)

Corrected.

P6, L183-184: repetition of vertical resolution

Corrected.

P8, L213: 01 -> 1

Corrected.

P11, L250: 77.4-90âŮę -> 77-90âŮę

We decided to stick to the notation of Pitts et al. (….), which is 77.8-90. Please note that the 77.4 has to read 77.8. This was a typo, which has been corrected.

Figure A1 caption: 1st -> 1

Corrected.

Figure 2-9, A1: The resolution of the figures is not good enough. On my printed version the plot frames are missing in several occasions.

Thanks for this hint. We did not experience any problems with the figures, but we will clarify this issue with the GMD production office.

Figure 4: Swap the upper panels with the lower ones, so that the order is July, August. Why do you use SREF as figure title? Why not using "SOCOL" as figure title?

We used $S_{REF}$ since the reference simulation (and not one of the sensitivity-runs) is shown. But it is absolutely correct, in the paper the Figure is shown before those sensitivity-simulations are introduced, so we changed the title. We further noticed that the plot were actually in the correct order (upper panel: July, lower panel: August), but the labels were swapped. This has been corrected. Thanks for pointing this out.

Figure 5: Here I would suggest to change the figure titles as follows: "CALIPSO", "SOCOL with thresholds", "SOCOL without thresholds".

Done.

Figures 7 and 8: Observations and model simulations are difficult to distinguish. I would suggest to use a thicker line for the observations and maybe a different color.

We revised Figures 7-9 and changed the line thickness and colors.

[Figure]

**Fig. 1.**

---

## Author Response (AR2)

**Author response to the topical editor decision: Publish subject to minor revisions (review by editor)**

We would like to thank the editor for his editor review and to identify the points that need minor revision.

We have taken all comments into account and revised the manuscript accordingly. A point-by-pont answer to the eight points are given below. These answeres are followed by a marked-up manuscript version.

1) Page 4, line 124 -- please do not use the term "SBS" -- as per my comments above, adding a new acronym for the sulphuric acid aerosol does not help the paper, and to me will serve only to further separate PSC research from other related science, in particular I am thinking about the stratospheric aerosol community here. Much better simply to state "also known as binary solution". Also the word "solutions" is odd, the word should be singular.
So my recommended revision here is to replace "(supercooled binary solution, SBS)" with "(also known as binary solution)".
We revised the section accordingly.

2) Line 136 -- replace "for the SBS particles" with "for the sulphuric acid aerosol particles".
We replaced the suggested text passage.

3) Line 144 -- when you state "mean radius" be precise about the type of mean radius -- I think you mean geometric mean radius here right (with a log-normal distribution and assumed geometric standard deviation)? If so please add "geometric" before "mean radius", and if a poly-disperse aerosol is assumed then add "(poly-disperse NAT particle size with log-normal distribution at geometric standard deviation of 1.x/2.x)" . Or if the calculation is for mono-disperse particle size, then just replace the word "mean" with "mono-disperse NAT particles of".
The calculation is for mono-disperse particle size. We changed the text accordingly.

4) Line 150 -- insert comma before "the PSC routine".
Done.

5) Line 153 -- insert "the" after "equilibrium between" and replace "phase" with "phases", deleting the hyphen between "gas" and "phase".
We adjusted the sentence accordingly.

6) Line 153 -- add "(i.e. the process is fast)" after "timescale is very short"
Added.

7) Lines 154-155 -- replace "the equilibrium between particulate and gas-phase is hardly reached, as shown by observations" with "the progression towards equilibrium of the particulate NAT phase and the gas phase is much slower, as inferred from observations". Replace also "and additional particles can nucleate" with "for example additional particles potentially nucleating"
We adjusted the sentence accordingly.

8) Line 178 -- This new sentence "All years are without volcanic influence" needs to be clarified. I think what you mean here is that the prescribed dataset providing the SAD of the binary solution is based on SAGE-II measurements that are outside periods of strong volcanism (e.g. during the unusual 1998-2002 period when the stratospheric aerosol layer was at background conditions). Assuming that is the case, then please add "(i.e. the prescribed SAD dataset representative of a background stratosphere, such as was the case in the 1998-2002 period)". However, if the CMIP6 dataset is used, then the dataset for the pre-industrial control is not a zero-volcanic background case, but more an average-volcanism case. Please consult with co-authors and re-word accordingly.

We used the CMIP6 data set, but not for the pre-industrial control. We used the CMIP6 data for the respective years of our study, namely 2006, 2007 and 2010. We meant that these years are not affected by extreme volcanic events like Mt. Pinatubo, but we agree that the stratospheric aerosol layer during these years is not as low as during 1998-2002. We reworded the text.

[revised manuscript text omitted]

---

## Author Response (AR3)

**Author response to the topical editor decision: Publish subject to technical corrections**

We thank the editor for his review and detailed wording suggestions, which further helped to improve our manuscript. We also thank the editor for recommending publication after these final changes. We further want to thank the two referees as well as Yunqian Zhu and Astrid Kerkweg for their helpful feedback and constructive comments. We appreciated the suggestions, which helped to improve the paper a lot.

We have re-worded the listed sentences and added the citations as suggested. A point-by-point replyis given below. A marked-up manuscript version is attached.

**Final minor wording changes (technical corrections)**
* * *
1) Page 9 line 235 -- the text "we compare SOCOL with CALIPSO along a single flight track" is not adequate. Firstly because CALIPSO is the name of the satellite, the name of lidar instrument being CALIOP. I appreciate the authors may prefer a writing style in brief sentences, but the wording needs to be precise when they do so. Also "flight track" could confuse some readers, the term "orbit" or "orbital transect" more appropriate for the satellite comparisons. Suggest to re-word to "we evaluate the vertical profile of SOCOL simulated PSCs, comparing to lidar measurements from the CALIPSO satellite, along specific orbital transects" or similar.

We changed the sentence accordingly.

2) Page 12, lines 314-316

These 3 sentences beginning "These values...", "However, the..." and "For example, the..." each need to be more specific

2.1) Suggest to change "These values had once been chosen based on what was known about PSCs back then." to "The values used for such PSC microphysics parameters tend to inherit early choices based on initial comparisons with limited observational datasets". Or if a particular study or choice was meant with the "back then" wording, then could instead explain that specifically, citing the paper the authors have in mind.

We reformulated the sentence.

2.2) Suggest to improve "However, the current parameter setting might not be optimal" to "Also, the initial parameter values chosen may reflect specific conditions in a particular polar winter, and may require adjustment to be more representative over the broader range of conditions experienced across several winters."

We changed the sentence as suggested.

2.3) With the text "For example, the rather low value of n_ice of 0.01 cm-3...", suggest to cite the study that provided the basis for this value, or the modelling study where this value was first chosen, if possible. Basically that sentence needs to be more specific as to why that value is now thought to be "rather low", whereas originally it was (presumably) a best-estimate.

The default setting for n_ice = 0.01 cm-3 was adopted from the previous model version SOCOL 2 (Schraner et al, ACP, 2008). Unfortunately, value of n_ice and its origin is not documented in that study. We found in the PhD thesis by Martin Schraner that in a first step n_ice was decreased from 10 cm-3 to 0.1 cm-3. This was done to decrease surface area densities and therefore heterogeneous reaction rates on ice. The second adjustment of n_ice to 0.01 cm-3 is not mentioned in the thesis. We assume that the estimate of 0.01 cm-3 was the result of internal discussions, probably taking simulated heterogeneous ozone loss into account.

To avoid any rating of the estimate for n_ice we removed the words "rather low". The point we want to make here is that with this setting the model cannot reproduce observed high number densities in mountain-wave clouds.

3) Page 16, lines 371-372 -- The sentence beginning "Both indicates a more efficient denitrification than in S$_{REF}$" needs to be improved. Suggest to add "features" after "Both", change "indicates" to "indicate" (grammar) and replace "a more efficient denitrification" with "greater denitrification". Suggest also adding "in the increased n$_{NAT,max}$ run (sensitivity run S$_{n(NAT,max)}$) )" before "than in S$_{REF}$".

The sentence has been rewritten as suggested.

4) Page 18, lines 404-406 -- The term "mimicking" is non-scientific, and does not really communicate adequately what is being done. The method is explained in the paper, and in my opinion it would be sufficient to just state "comparing to the space borne lidar measurement" rather than "mimicking a lidar measurement" (also changing "compared modelled" earlier in the sentence instead to "evaluated simulated" to avoid duplication of "compared"). If the authors want to communicate some specifics about their method for doing that, they should choose a better word than "mimicking". Maybe reword "mimicking a lidar measurement on the model output" to "deriving an equivalent backscatter metric from the model, along CALIPSO orbital transects, and aligning with optical thresholds used in the CALIOP type algorithm" or similar.

We have changed the sentence to "We evaluated modeled PSC occurrence and composition to CALIPSO/CALIOP satellite observations by deriving an equivalent backscatter metric from the model output and aligning with optical thresholds used in the CALIOP classification algorithm."

5) Page 18, lines 410-412. The wording in these 2 sentences need to be improved

5.1) suggest adding "In the model, " at the start of the sentence and replacing "without taking" with ",

whereas NAT particles are known to require several days to grow to larger sizes, their size then being dependent on..". I'm not sure why the authors have added "the pre-existing PSC distribution", although I guess maybe they mean the sequestering of HNO3 into liquid STS particles. If that was what was meant then I think that effect would already be resolved in the current PSC scheme (even with the simplified NAT approach). In which case I suggest to delete "and pre-existing PSC distributions into account" and cite a paper that has shown the effect explained re: the impact of resolving the time required for growth compared to a PSC scheme with instantaneous NAT formation, for example Feng et al. (2011, ACP).

We therefore changed the sentence as suggested and added cited the paper of Feng et al. (2011) in the following sentence (also reformulated according to 5.2):

"In the model, NAT and ice PSCs form instantaneously, whereas NAT particles are known to require several days to grow to larger sizes, their size then being dependent on the history of the air mass. The instantaneous NAT formation approach therefore represents a simplification, but is successfully applied in several other models (e.g., Wegner et al., 2013) and has been evaluated against PSC schemes resolving the time required for growth (e.g., Feng et al., 2011).

5.2) The 2nd sentence, which begins "The approach" again needs to be more specific -- suggest to re-word to "The instantaneous NAT formation approach". Also, the word "states" is not appropriate -- suggest to replace to "therefore represents". The latter part of the sentence can also be improved, inserting "several" before "other models" and deleting "as well".

We reformulated the sentence, see 5.1.

6) Page 18, lines 413-419 -- these sentences were poorly worded in places.

6.1) Lines 413-414 -- Please insert "simplified" before "PSCs scheme" improve the wording of "includes several fixed microphysical parameters" -- suggest to re-word to "then requires to assign representative constant values for several PSC microphysics parameters." or similar wording.

We reworded the sentence as suggested.

6.2) Lines 415-416 -- Please also re-word "The actual value for the NAT number density" because there is no single value for that -- it varies depending on the situation (e.g. the recirculation of air within the cold region or variations in the concentration of preferential-nuclei, e.g. meteoric smoke particles). This potential reasons for the variation do not need to be specified explicitly here -- but this sentence here is clearly trying to communicate that the constant value is a simplification, the situation much more variable. In which case, suggest to re-word by deleting "the actual value for" and "being". The reference to the active sites is referring to active sites on the NAT-freezing behaviour of aerosol particles containing preferential nuclei (e.g. meteoric particle) and seems too detailed for that 2nd part of the sentence. Suggest to first add after "because" the text "of different cold-pool and vortex situation (e.g. Mann et al., 2003), the availability of NAT nuclei (Voigt et al., 2005)". With that re-wording, then the "active sites" can be a continuation of the NAT nuclei point, changing "NAT nucleation themselves

shows" instead to "NAT nuclei themselves showing". In addition to the Hoyle et al. (2013), suggest to also cite the James et al. (2018) ACP paper, which progresses this discussion re: the nature of the preferential NAT nuclei.

We changed the sentence to: " In reality, the NAT number density is far from constant, because of different cold-pool and vortex situations (e.g., Mann et al., 2003), the availability of NAT nuclei (Voigt et al., 2005) themselves showing a wide distribution of efficacies (Hoyle et al., 2013; James et al., 2018)."

6.3) Line 418 -- Suggest to replace "tuning of the microphysical parameters to reasonably" instead to "testing to reach representative microphysical parameter values which reasonably"

Done.

6.4) Line 419 -- The sentence beginning "This was done here by various" similarly needs to be improved. Please replace those 6 words with "In this study, we have explored the implications of some of the values chosen in the simplified PSC scheme, analysing several". Then adding the following text before the full stop ", progressing to new values which we show perform reasonably well across the broader conditions found across several Antarctic winters" or similar.

We changed and extended the sentence as suggested.

7) Lines 462-468 -- again, this part of the text still requires improved wording.

7.1) Lines 462-464 -- the assertion "we do not expect any substantial drawbacks from the applied set-up" needs to more clearly be communicated to be specific to the "specified-dynamics" (i.e. nudged) situation. This can be done simply by replacing "the applied set-up. First the model was used" to "the applied set-up, provided the model is used" and change the comma after "dynamics mode" instead to a full stop.

Done.

7.2) Lines 464-465 -- With that re-wording in 7.1) above, please change "and we do not see major differences" to "The nudged approach ensures no significant differences" and delete "in nudged mode" at the end of the sentence.

Done.

7.3) The wording "Second, to capture mountain wave events, for example, we would need to go to" seems too sudden to discuss this without citing a reference -- suggest to re-word that text instead to "Although mountain waves are known to be important in triggering NAT nucleation (e.g. Carslaw et al., 1999), resolving such localised PSC formation explicitly would require", deleting "which are" and replacing "beyond the capabilities of" with "inconsistent with the computational constraints of".

We reworded the sentence and added the citation as suggested.

7.4) Lines 466-468 -- Again the wording of this sentence should be improved -- suggest to change

"This is supported by Khosrawi et al. (2017), who found only little differences" with "Also, Khosrawi et al. (2017) found only small differences", deleting also the "a" before "T42" and "T106", and adding "simulations of the cold 2015/6 Arctic winter" at the end of the sentence (to be clear that is for that particular winter.

We followed the suggestion.

**Minor additional typo-/grammatical-improvements.**
* * *
1) Page 18, line 403 -- insert "have" before "presented an evaluation".

Done.

**Papers**
* * *
Carslaw, K. S., Peter, T., Bacmeister, J. T. and Eckermann, S. D.
"Widespread solid particle formation by mountain waves in the Arctic stratosphere", J. Geophys. Res., vol. 104, no. D1, pp. 1827-1836 (1999).

Feng, W., Chipperfield, M. P., Davies, S., Mann, G. W., Carslaw, K. S. et al.
"Modelling the effect of denitrification on polar ozone depletion for the Arctic winter 2004/5", Atmos. Chem. Phys., vol. 11, pp. 6,559-6,573 (2011).

James, A. D., Brooke, J. S. A., Mangan, T. P., Whale, T. F., Plane, J. M. C. and Murray, B. J.: "Nucleation of nitric acid hydrates in polar stratospheric clouds by meteoric material", Atmos. Chem. Phys., vol. 18, pp. 4519-4531. (2018)

Mann, G. W., Davies, S., Carslaw, K. S. and Chipperfield, M. P.
"Factors controlling Arctic denitrification in cold winters of the 1990s"
Atmos. Chem. Phys., vol. 3, pp. 403-416, (2003)

[revised manuscript text omitted]
  features indicate greater denitrification in the increased $n_{NAT,max}$ run (sensitivity run $S_{n(NAT,max)}$)) than in $S_{REF}$.

The simulation $S_{n(ice),n(NAT,max)}$ (magenta), in which $n_{NAT,max}$ is twice as large as in $S_{REF}$, but only half of $S_{n(NAT,max)}$, falls in between the other simulations. The denitrification starts about half a month later than in $S_{n(NAT,max)}$. The $HNO_3$-uptake is reduced and subsequently $HNO_3$ stays longer in the gas-phase. However, in August $HNO_3$ concentrations reach about the same level as in $S_{n(NAT,max)}$. Simulations with enhanced $r_{NAT}$ have similar effects (not shown).

In $S_{T,n(ice),n(NAT,max)}$ denitrification as well as renitrification are delayed by about half a month due to the later onset of PSC formation. However, towards the end of the winter, $HNO_3$ concentrations are almost the same in all model simulations.

[Figure]

**Figure 8.** Same as Fig. 7, but for $H_2O$. Note that the line of $S_{n(NAT,max)}$ overlays $S_{REF}$, since these simulations have identical $H_2O$.

Figure 8 shows the same as Fig. 7, but for $H_2O$. As for $HNO_3$, all simulations start with similar $H_2O$ values in May, but underestimate MLS by 20% to 30%. At 46 hPa MLS $H_2O$ starts to decline beginning of June. Rehydration of lower levels due to the evaporation of sedimenting ice particles is observed shortly after. At 68 hPa, MLS $H_2O$ starts to decrease mid
385 of June. All model simulations except for $S_{T,n(ice),n(NAT,max)}$ show a very similar temporal evolution of $H_2O$ in the polar stratosphere and a very good agreement with MLS. In SOCOL the amount of ice is determined by the amount of available $H_2O$ and temperatures. The smaller the chosen $n_{ice}$, the larger the ice particles and the stronger the dehydration due to faster sedimentation. $S_{REF}$ and $S_{n(NAT,max)}$, the simulations with the lowest $n_{ice}$ of $0.01$ cm$^{-3}$, show the strongest dehydration and the earliest onset, while $S_{n(ice)}$ with $n_{ice} = 0.1$ cm$^{-3}$ shows the smallest dehydration of the simulations without modified
390 PSC formation temperature. With the cold bias correction of +3 K, almost no dehydration takes place due to lack of ice formation. Changes in polar vortex $H_2O$ from modifying $n_{ice}$ have an influence on the SAD of NAT and STS, with higher $H_2O$ concentrations leading to larger NAT and STS SADs. However, this effect is small compared to the effects from modifying the NAT-related microphysical parameters, and therefore, not further discussed.

Finally, Fig. 9 presents simulated $O_3$ in the polar stratosphere compared to MLS. At the beginning of winter all model
395 simulations are in very good agreement with MLS measurements. For both pressure levels, the simulations show an earlier and stronger decline in $O_3$ than observed by MLS. Also, the recovery of $O_3$ starts earlier, leading to slightly higher $O_3$ values at the end of October. The spread among the model simulations is small compared to the differences to the observations, indicating minor effects of the PSC parameters on $O_3$-depletion. Increasing the parameter $n_{ice}$ slightly reduces the simulated dehydration, but the increased SAD of ice leads to a slightly stronger $O_3$ depletion in $S_{n(ice)}$ compared to $S_{REF}$. Allowing for higher

[Figure]

**Figure 9.** Same as Fig. 7, but for $O_3$.

NAT number densities overall reduces SAD of PSCs due to reducing the abundance of $HNO_3$. However, due to enhanced denitrification, $S_{n(NAT,max)}$ and $S_{n(ice),n(NAT,max)}$ show even slightly lower $O_3$ concentrations. $O_3$-depletion starts later in $S_{T,n(ice),n(NAT,max)}$ due to the later onset of PSC occurrence and smaller PSC area. However, from end of August onwards the differences between the individual model simulations vanish. The discussed findings for $HNO_3$, $H_2O$ and $O_3$ hold also for the years 2006 and 2010, as shown in Figs. A11 to A14.

**4  Discussion and Conclusions**

We have presented an evaluation of polar stratospheric clouds (PSCs) in the Antarctic stratosphere as simulated by the chemistry-climate model SOCOLv3.1. The model was nudged towards ERA-Interim reanalysis (specified dynamics mode). We  evaluated modeled PSC occurrence and composition to CALIPSO/CALIOP satellite observations by  deriving an equivalent backscatter metric from the model output and aligning with optical thresholds used in the CALIOP classification algorithm. The impact of PSCs on the chemical composition of the polar stratosphere by denitrification, dehydration and ozone depletion was investigated by comparison with Aura/MLS satellite data. We analysed three winters with different PSC occurrence: 2006 (above-average), 2007 (average) and 2010 (below-average).

SOCOL considers STS droplets as well as water ice and NAT particles. PSCs are parametrized in terms of temperature and partial pressures of $HNO_3$ and $H_2O$, assuming equilibrium conditions. In the model, NAT and ice PSCs form instantaneously , whereas NAT particles are known to require several days to grow to larger sizes, their size then

being dependent on the history of the air mass . The instantaneous NAT formation approach therefore represents a simplification, but is successfully applied in  several other models (e.g., Wegner et al., 2013) and has been evaluated against PSC schemes resolving the time required for growth (e.g., Feng et al., 2011).

420   The  simplified PSCs scheme then requires to assign representative constant values for several PSC microphysics parameters, namely the maximum NAT number density, NAT radius and ice number density. Fixing the NAT radius leads to a homogeneous sedimentation velocity for all NAT particles, but allows for varying NAT number densities. Other models choose the reverse approach with fixed number densities, which results in varying NAT radius and sedimentation velocities (e.g., Wegner et al., 2013; Nakajima et al., 2016). In reality, the

425    NAT number density is far from  constant, because  of different cold-pool and vortex situations (e.g., Mann et al., 2003), the availability of NAT nuclei (Voigt et al., 2005) themselves showing a wide distribution of efficacies (Hoyle et al., 2013; James et al., 2018). Both approaches require some  testing to reach representative microphysical parameter values which reasonably simulate observed sedimentation and denitrification. In this study,

430   we have explored the implications of some of the values chosen in the simplified PSC scheme, analysing several sensitivity simulations, progressing to new values which we show perform reasonably well across the broader conditions found across several Antarctic winters.

[revised manuscript text omitted]

James, A. D., Brooke, J. S. A., Mangan, T. P., Whale, T. F., Plane, J. M. C., and Murray, B. J.: Nucleation of nitric acid hydrates in polar

590    stratospheric clouds by meteoric material, Atmospheric Chemistry and Physics, 18, 4519–4531, https://doi.org/10.5194/acp-18-4519-2018, https://acp.copernicus.org/articles/18/4519/2018/, 2018.

Jiang, Y. B., Froidevaux, L., Lambert, A., Livesey, N. J., Read, W. G., Waters, J. W., Bojkov, B., Leblanc, T., McDermid, I. S., Godin-Beekmann, S., Filipiak, M. J., Harwood, R. S., Fuller, R. A., Daffer, W. H., Drouin, B. J., Cofield, R. E., Cuddy, D. T., Jarnot, R. F., Knosp, B. W., Perun, V. S., Schwartz, M. J., Snyder, W. V., Stek, P. C., Thurstans, R. P., Wagner, P. A., Allaart, M., Andersen, S. B., Bodeker, G., Calpini, B., Claude, H., Coetzee, G., Davies, J., De Backer, H., Dier, H., Fujiwara, M., Johnson, B., Kelder, H., Leme, N. P., Konig-Langlo, G., Kyro, E., Laneve, G., Fook, L. S., Merrill, J., Morris, G., Newchurch, M., Oltmans, S., Parrondos, M. C., Posny, F., Schmidlin, F., Skrivankova, P., Stubi, R., Tarasick, D., Thompson, A., Thouret, V., Viatte, P., Vömel, H., von der Gathen, P., Yela, M., and Zablocki, G.: Validation of Aura Microwave Limb Sounder ozone by ozonesonde and lidar measurements, J. Geophys. Res., 112, https://doi.org/10.1029/2007jd008776, 2007.

Jourdain, L., Bekki, S., Lott, F., and Lefevre, F.: The coupled chemistry-climate model LMDz-REPROBUS: description and evaluation of a transient simulation of the period 1980-1999, Ann. Geophys., 26, 1391–1413, https://doi.org/10.5194/angeo-26-1391-2008, 2008.

Khosrawi, F., Kirner, O., Sinnhuber, B.-M., Johansson, S., Höpfner, M., Santee, M. L., Froidevaux, L., Ungermann, J., Ruhnke, R., Woiwode, W., Oelhaf, H., and Braesicke, P.: Denitrification, dehydration and ozone loss during the 2015/2016 Arctic winter, Atmospheric Chemistry and Physics, 17, 12 893–12 910, https://doi.org/10.5194/acp-17-12893-2017, https://acp.copernicus.org/articles/17/12893/2017/, 2017.

Khosrawi, F., Kirner, O., Stiller, G., Höpfner, M., Santee, M. L., Kellmann, S., and Braesicke, P.: Comparison of ECHAM5/MESSy Atmospheric Chemistry (EMAC) simulations of the Arctic winter 2009/2010 and 2010/2011 with Envisat/MIPAS and Aura/MLS observations, Atmos. Chem. Phys., 18, 8873–8892, https://doi.org/10.5194/acp-18-8873-2018, 2018.

Kirner, O., Müller, R., Ruhnke, R., and Fischer, H.: Contribution of liquid, NAT and ice particles to chlorine activation and ozone depletion in Antarctic winter and spring, Atmos. Chem. Phys., 15, 2019–2030, https://doi.org/10.5194/acp-15-2019-2015, 2015.

Lambert, A., Read, W. G., Livesey, N. J., Santee, M. L., Manney, G. L., Froidevaux, L., Wu, D. L., Schwartz, M. J., Pumphrey, H. C., Jimenez, C., Nedoluha, G. E., Cofield, R. E., Cuddy, D. T., Daffer, W. H., Drouin, B. J., Fuller, R. A., Jarnot, R. F., Knosp, B. W., Pickett, H. M., Perun, V. S., Snyder, W. V., Stek, P. C., Thurstans, R. P., Wagner, P. A., Waters, J. W., Jucks, K. W., Toon, G. C., Stachnik, R. A., Bernath, P. F., Boone, C. D., Walker, K. A., Urban, J., Murtagh, D., Elkins, J. W., and Atlas, E.: Validation of the Aura Microwave Limb Sounder middle atmosphere water vapor and nitrous oxide measurements, J. Geophys. Res., 112, https://doi.org/10.1029/2007jd008724, 2007.

Livesey, N. J., Read, W. G., Wagner, P. A., Froidevaux, L., Lambert, A., Manney, G. L., Millán Valle, L. F., Pumphrey, H. C., Santee, M. L., Schwartz, M. J., Wang, S., Fuller, R. A., Jarnot, R. F., Knosp, B. W., Martinez, E., and Lay, R. R.: Earth Observing System ( EOS ), Aura Microwave Limb Sounder ( MLS ), Version 4.2 Level 2 data quality and description document, https://mls.jpl.nasa.gov/data/v4-2_data_quality_document.pdf, 2018.

Mann, G. W., Davies, S., Carslaw, K. S., and Chipperfield, M. P.: Factors controlling Arctic denitrification in cold winters of the 1990s, 
[revised manuscript text omitted]

Voigt, C., Schlager, H., Luo, B. P., Dörnbrack, A., Roiger, A., Stock, P., Curtius, J., Vössing, H., Borrmann, S., Davies, S., Konopka, P., Schiller, C., Shur, G., and Peter, T.: Nitric Acid Trihydrate (NAT) formation at low NAT supersaturation in Polar Stratospheric Clouds (PSCs), Atmospheric Chemistry and Physics, 5, 1371–1380, https://doi.org/10.5194/acp-5-1371-2005, https://acp.copernicus.org/articles/

695  5/1371/2005/, 2005.

Waters, J. W., Froidevaux, L., Harwood, R. S., Jarnot, R. F., Pickett, H. M., Read, W. G., Siegel, P. H., Cofield, R. E., Filipiak, M. J., Flower, D. A., Holden, J. R., Lau, G. K. K., Livesey, N. J., Manney, G. L., Pumphrey, H. C., Santee, M. L., Wu, D. L., Cuddy, D. T., Lay, R. R., Loo, M. S., Perun, V. S., Schwartz, M. J., Stek, P. C., Thurstans, R. P., Boyles, M. A., Chandra, K. M., Chavez, M. C., Chen, G. S., Chudasama, B. V., Dodge, R., Fuller, R. A., Girard, M. A., Jiang, J. H., Jiang, Y. B., Knosp, B. W., LaBelle, R. C., Lam, J. C., Lee, K. A.,

700  Miller, D., Oswald, J. E., Patel, N. C., Pukala, D. M., Quintero, O., Scaff, D. M., Van Snyder, W., Tope, M. C., Wagner, P. A., and Walch, M. J.: The Earth Observing System Microwave Limb Sounder (EOS MLS) on the Aura satellite, IEEE Trans. Geosci. Remote Sens., 44, 1075–1092, https://doi.org/10.1109/Tgrs.2006.873771, 2006.

Wegner, T., Grooss, J. U., von Hobe, M., Stroh, F., Suminska-Ebersoldt, O., Volk, C. M., Hosen, E., Mitev, V., Shur, G., and Muller, R.: Heterogeneous chlorine activation on stratospheric aerosols and clouds in the Arctic polar vortex, Atmospheric Chemistry and Physics, 12, 11 095–11 106, https://doi.org/10.5194/acp-12-11095-2012, <GotoISI>://WOS:000312411300030, 2012.

Wegner, T., Kinnison, D. E., Garcia, R. R., and Solomon, S.: Simulation of polar stratospheric clouds in the specified dynamics version of the whole atmosphere community climate model, J. Geophys. Res., 118, 4991–5002, https://doi.org/10.1002/jgrd.50415, 2013.

Winker, D. M. and Pelon, J.: The CALIPSO mission, in: IEEE International Geoscience and Remote Sensing Symposium. Proceedings, vol. 2, pp. 1329–1331, https://doi.org/10.1109/IGARSS.2003.1294098, 2003.

Winker, D. M., Hunt, W. H., and McGill, M. J.: Initial performance assessment of CALIOP, Geophys. Res. Lett., 34, https://doi.org/10.1029/2007gl030135, 2007.

Winker, D. M., Vaughan, M. A., Omar, A., Hu, Y. X., Powell, K. A., Liu, Z. Y., Hunt, W. H., and Young, S. A.: Overview of the CALIPSO Mission and CALIOP Data Processing Algorithms, J. Atmos. Oceanic Technol., 26, 2310–2323, https://doi.org/10.1175/2009jtecha1281.1, 2009.

Wohltmann, I., Lehmann, R., and Rex, M.: The Lagrangian chemistry and transport model ATLAS: simulation and validation of stratospheric chemistry and ozone loss in the winter 1999/2000, Geosc. Model Dev., 3, 585–601, https://doi.org/10.5194/gmd-3-585-2010, 2010.

Zhu, Y. Q., Toon, O. B., Lambert, A., Kinnison, D. E., Bardeen, C., and Pitts, M. C.: Development of a Polar Stratospheric Cloud Model Within the Community Earth System Model: Assessment of 2010 Antarctic Winter, Journal of Geophysical Research-Atmospheres, 122, 10 503–10 523, https://doi.org/10.1002/2017jd027003, <GotoISI>://WOS:000413675900022, 2017a.

Zhu, Y. Q., Toon, O. B., Pitts, M. C., Lambert, A., Bardeen, C., and Kinnison, D. E.: Comparing simulated PSC optical properties with CALIPSO observations during the 2010 Antarctic winter, J. Geophys. Res., 122, 1175–1202, https://doi.org/10.1002/2016jd025191, 2017b.